# A Literature Review of Incorporating Crack Tip Plasticity into Fatigue Crack Growth Models

**DOI:** 10.3390/ma16247603

**Published:** 2023-12-11

**Authors:** Antonio Garcia-Gonzalez, Jose A. Aguilera, Pablo M. Cerezo, Cristina Castro-Egler, Pablo Lopez-Crespo

**Affiliations:** Department of Civil and Materials Engineering, University of Malaga, C/Dr Ortiz Ramos, S/N, 29071 Malaga, Spain; j.a.aguilera.garcia@uma.es (J.A.A.); pm@uma.es (P.M.C.); kcegler@gmail.com (C.C.-E.); plopezcrespo@uma.es (P.L.-C.)

**Keywords:** stress intensity factor (SIF), linear elastic fracture mechanics (LEFM), linear elasto–plastic fracture mechanics (LEPFM), crack closure (CC)

## Abstract

This paper presents an extensive literature review focusing on the utilisation of crack tip plasticity as a crucial parameter in determining and enhancing crack growth models. The review encompasses a comprehensive analysis of various methodologies, predominantly emphasising numerical simulations of crack growth models while also considering analytical approaches. Although experimental investigations are not the focus of this review, their relevance and interplay with numerical and analytical methods are acknowledged. The paper critically examines these methodologies, providing insights into their advantages and limitations. Ultimately, this review aims to offer a holistic understanding of the role of crack tip plasticity in the development of effective crack growth models, highlighting the synergies and gaps between theoretical, experimental, and simulation-based approaches.

## 1. Introduction

During the last third of the last century and so far this century, the publication of reviews on crack growth models has increased considerably. Fatigue failure has been a widely researched topic in academia over the past few years, and has been discussed in several publications that cite key contributions from researchers such as Bathias and Baïlon [1] or Suresh [2]. Crack closure and crack opening models have been discussed [3], focusing on the volume-based strain energy density approach applied to welded structures with notches [4] and the orientation of the critical plane in multiaxial fatigue [5]. Pippan and Hohenwarter presented a review on the controversial phenomenon of crack closure and its physical implications and consequences [6], a study focusing on the theory of critical distances [7], reviews of the application of the digital image correlation (DIC) technique to fatigue [8,9] and reviews of the use of the finite element method (FEM) in the study of crack growth [10,11,12]. In another compilation work [13] on notched components, different types of approaches have been presented to characterise notch fatigue behaviours under complex loading histories. Among them, the consideration of the stress gradient effect is a popular perspective, together with the concepts of fatigue damage zone and critical distance.

Fatigue crack growth has been extensively studied using the curves da/dN—ΔK. ΔK is essentially an elastic parameter, while the fatigue crack growth process is related to non-linear and irreversible plastic phenomena at the crack edge. This explains the limitation of da/dN—ΔK models in predicting the effect of stress ratio or load amplitude variation. Among the limitations are: (a) such curves are completely phenomenological, not derived from physics, and the fitting parameters have no physical justification; (b) such curves are only valid in the small-scale yield range; and (c) da/dN depends on other parameters, including stress ratio and loading history.

Fatigue damage is reflected in plastic phenomena that are not analysed by elastic models, so it is necessary to introduce different mechanisms to analyse the damage that occurs in each cycle. One of these is the cyclic plastic deformation, in which the damage accumulation process is strongly influenced by the evolution of the dislocation substructure [14]. The presence of surface oxides in the base material can act as crack initiation points and can significantly decrease the fatigue life of the material [15]. They can also act as brittle inclusions and behave as cracks when loaded normal to their long axes [16].

Brittle solids often exhibit similar micro mechanisms for the growth of static and cyclic cracks at low temperatures [2]. However, the application of a cyclic load on a static mean stress can result in variable lifetimes and, in some cases, the cyclic frequency can influence the crack growth rate. Furthermore, under fully compressive cyclic loading, crack initiation can occur solely due to cyclic loading, even in regions without stress concentrations. These phenomena can significantly influence the growth and failure of brittle solids. Another failure mechanism is ‘Fatigue micro voids’, which occur because of the separation of non-coherent secondary particles from the matrix [17]. The void formation process is characterised by the initiation, growth, and coalescence of multiple interfacial cracks around the particle.

Subsequently, the phenomenon of crack closure is introduced, which was proposed by [18] and has been used to explain the influence of plasticity on fatigue crack growth. It has also been used to determine numerically, experimentally, and analytically the effect of stress ratio [19], short cracks [20,21], and model widths [22]. From here, the first in-depth studies [23] evolved to elucidating the concept of partial crack closure [24]. Following this, ref. [25] proposed a new model for the stress around the crack based on four parameters: KF (SIF opening mode), KS (SIF shear), KR (SIF retardation), and T-strain. KR characterises the effect of crack tip protection arising due to plasticity in both the crack tip and the wake. The use of non-linear parameters was introduced to validate the concept of crack closure and identify the most accurate parameters [26]. Experimentally, a relationship between crack closure and crack growth has been observed with different techniques; in [27], DIC is used for this, and subsequently, a model was established to reproduce the results, obtaining information regarding the closing stresses. In another study [28], a direct relationship between overloads, crack closure, and how this affects crack growth is observed. In [29,30], a relationship is established between plastic wake sizes, crack opening displacement (COD), and effective intensity factor at the crack tip. Following that, these parameters have also been related to crack closure with DIC experimental techniques [31]. The DIC technique has been shown to be a very useful tool for determining the elastic or elasto–plastic displacement field in the crack environment, to later, through mathematical, analytical, or numerical models, calculate the key parameters of the fatigue crack growth rate (FCGR), and establish the possible driving forces of crack growth. Only in [32] do the authors evaluate five different methods to locate the crack tip position by applying DIC. The methods are: the two constrained Newton; the trust–region reflective and the quasi-Newton; the Nelder–Mead Simplex; a constrained genetic algorithm (GA), and finally a constrained pattern search (PS).

The X-DIC technique [33] has also been shown to be efficient in evaluating the plastic state of the crack tip. Today, there is still a lot of controversy surrounding the crack closure phenomenon. On the one hand, many researchers deny that it has any influence on crack growth; on the other hand, Pippian and Grosinger [34] proposed an original approach to the crack closure phenomenon, extending its importance and relevance. At the time of the research publication, the acceptance of the crack closure phenomenon and its effect on the FCGR was restricted to low cycle fatigue (LCF) processes. Based on images from a scanning electron microscope (Figure 1c) and the interpretation of ΔJ and ΔJ_eff_ in the cyclic stress–strain hysteresis curves characteristic of fatigue processes (Figure 1a,b), the authors demonstrated that crack closure also occurs in high cycle fatigue (HCF) processes and affects ΔJ and ΔJ_eff_, consequently affecting the FCGR.

The study of the effects of elasto–plastic behaviour on crack growth has been approached from many points of view and mechanisms. For example, the effect of the sharpening and hardening that occurs at the crack tip on the crack development itself has been addressed [35]. The effect of crack tip sharpening and hardening under maximum and minimum loads has been investigated [36]. This phenomenon has also been experimentally verified [37] with the use of micro fractography. A mathematical model was proposed [38] based on time differential equations describing the thermodynamic process of plastic energy dissipation during the hardening and sharpening of the crack tip growth. Another analytical approach [39] is based on the use of the Smith–Watson–Topper parameter, assuming that in the first loading cycles a large and irreversible sharpening of the crack tip occurs.

Another important element to consider is the identification of the crack tip plastic zone. During fatigue crack propagation, two plastic zones are created at the crack tip [40]. One is constant during loading, while the other is cyclic during unloading. Research has shown that the size and shape of the plastic zone significantly influences crack growth behaviour.

It should also be mentioned that there are numerous papers whose focus is not on the relationship between crack growth and crack tip plasticity, but which provide evidence of the connection. In [41], whose main objective was to validate crack growth models (Fatemi–Socie) under biaxial loads and overloads, it was clearly observed that the combined application of axial and torsion almost eliminated crack closure, thereby increasing the crack growth rate. In the following work [42], a finite element model was analysed with Abaqus and the effect of the position of holes drilled in a plate to reduce the growth rate of future cracks was evaluated. The optimum location and size were sought, reducing the rate of crack growth by 50%. To see the effect of the circular holes drilled in the crack tip region, a direct evaluation of the ratchet limit and the plastic deformation range of the crack tip was made. This evaluation was achieved using the new linear matching method (LMM). There is a situation that optimises the slowing down of crack growth because the plastic deformation decays by 50%.

This article systematically reviews recent advances in notch fatigue analysis in relation to stress gradient effects. Despite these advances, a comprehensive analysis of published methods that use crack tip plasticity to determine or improve crack growth models has not yet been carried out. Next, different research that proposes methods for determining crack growth is presented. Works that establish quantifiable relationships between plasticity and crack growth are also presented. Although they do not present a complete model, they detect and quantify parameters of interest and are the seed of possible future models when all the parameters are found, which, as they are unknown, do not allow the formulation of the complete model. The works have been grouped according to the theories used, as well as the damage parameters. The organization of the review sections is based on the chronological order of the publication year of the works under examination. Additionally, within each section, the presented research is arranged in chronological order to facilitate the comprehension of each model’s timeline.

## 2. Cumulative Damage Theories Based on Stress and/or Strain History

Building upon the foundational theories in the field of crack growth, a variety of models have been proposed, each extending and refining the understanding of the underlying mechanisms. These models, though varied, share a common lineage in their approach to the study of cumulative damage based on the stress and strain history in fatigue crack growth.

Focusing on the primary aspects of cumulative damage theories, one of the earliest works was proposed by Shih [43]. This work presents a relation between the J-integral and the crack tip opening displacement (CTOD) obtained, based on the description of the stress and strain state proposed by Hutchinson–Rice–Rosengren with a singularity at the crack tip. A similar expression is formulated for the CTOD. The author arrives at a relationship between CTOD and J in the equations. The coefficient *d*, which relates to *J*, depends on the deformation properties of the material and is independent under small-scale plasticisation conditions. However, the obtained relations are not valid for large-scale plasticisation due to the singularity. Based on deformation theory, the author arrives at a relation between *J* and the parameter *d_n_*, Equation (1), which is a candidate for a more robust model. In fact, after numerical correlations, the model is found to be good at all scales.
(1)$$d\delta /da=\left(\frac{n}{n+1}\right)\frac{d_{n}}{\sigma_{0}}\frac{dJ}{da}$$

Building on this foundation, Noroozi et al. [39] introduced a new perspective by presenting a FCGR model that hinges on the elasto–plastic stress–strain history at the crack tip. The driving force of crack development was derived based on the damage parameter used by Smith–Watson–Topper. The internal effect of stresses induced by the inverse plasticity cycle was also analysed. All these derive to Equation (2), where the final model is presented as a function of the two parameters and where the authors present an extensive network of mathematical developments and balances of forces and stresses. Evidently, the complexity of this model lies in expressing this history of stresses and strains, so the focus is on the maximum and minimum stresses and strains at the crack tip (in Equations (8)–(16) in the original publication). Special attention is paid to the contribution of compressive stresses (in Equations (17)–(24) in the original publication), while elasto–plastic stresses and strains in the crack tip environment are described (in Equations (27)–(32) in the original publication). The calculation of the residual intensity factor K_r_ is addressed (in Equations (33) and (34) in the original publication) and all other necessary intensity factors are listed (in Equations (35)–(40) in the original publication). With all the above, the fatigue crack driving force ΔK and the fatigue crack growth expression *da*/*dN − ΔK* are derived analytically (in Equations (41)–(60) in the original publication). Figure 2 shows a high correlation of the model results with the experimental ones. The authors point out that the fact that being able to relate the Smith–Watson–Topper damage parameter and the stress-strain history of the crack allowed introducing residual stresses into the model has improved its goodness of fit.
(2)$$da/dN={{f((K_{max,tot})}^{\rho }{\left(K_{tot}\right)}^{1-\rho })}^{-\gamma }$$

In a similar vein, de Matos and Nowell [44] extended previous models by integrating the plane strain plastic-induced crack closures (PICC) model for more complex geometries, such as a flat plate with a circular hole and two radially symmetric cracks, and residual stress fields. This model is an improvement because it can be extended to any geometry and previous residual stress fields. The analytical model starts by expressing the stress state as well as possible residual stresses with boundary elements and plane stress, while the objective function is set as the crack growth. A finite element model is implemented and solved for both elastic and elasto–plastic cracks, with and without residual stresses. The results are presented in Figure 3, which correlates the results of the analytical models with the experimental.

Extending the exploration of cumulative damage theories to the dynamic interplay between plastic zones and crack growth, Chikh et al. [40] presented a study focusing on this relationship. Since two plastic zones are produced under cyclic loading, one monotonic (*r_m_*) and one cyclic (*r_c_*), this work relates this second zone *r_c_* to the FCGR, which is produced by the local compression generated at the crack closure of each cycle. Several models can be used to relate the plastic zone to crack propagation. In this study, a simplified model based on the effect of damage accumulated in front of the crack tip *da*/*dN = f(r_c_)* is used. Taking the expression of the stress value around the linear elastic fracture mechanics (LEFM) crack, it implies that a plastic deformation zone is formed around the crack which has an extension relative to the value of the applied load and the crack length. Furthermore, if the plastic zone is large, a large amount of energy is absorbed during crack propagation, and if the zone is small, crack propagation requires less energy; thus, the size of the plastic zone is directly proportional to the hardness of the material. Therefore, the mechanical properties of the material and the stress state govern the size and shape of the plastic zone. It is concluded that the plastic zone depends on the stress field, the applied stress, the specimen thickness, and the crack length, and that the crack size is an ideal fatigue crack propagation parameter to determine crack growth, better than any other elasto–plastic fracture mechanics (EPFM) parameter. The proposed model is based on *da*/*dN = B(r_c_) ^2^*, where B = 3.2 × 10^−6^ (R_e_)^1.3^, R_e_ being the yield stress. Experimental tests were carried out on 12NC6 steel, yielding consistent and conservative results.

In the context of high-temperature conditions, Tong et al. [45,46] bring a novel perspective by incorporating finite element numerical modelling. They also present further results from finite element numerical modelling at high temperatures (650 °C). The modelling of the material follows Chaboche’s laws [47] and isotropic kinematic hardening (Power Law), and the results are shown after the application of 30 loading cycles under different values of load, frequency, and load ratios (*R*). Among the main conclusions, (a) the ratchetting deformation increases as the *R* decreases, being higher in negative cases, (b) low frequencies in the loading cycle increase the accumulated deformation, as well as the crack growth rate, and (c) maintaining an initial load (Dwell) decreases the fatigue life of the specimen; for example, a Dwell of 100 s at maximum initial load reduces the fatigue life by one third. An interesting factor studied [45] is the use of meshing to define different grain sizes and orientations. It is important to mention the study from Lee et al. [48], where a method of extending the use of the *Ct* parameter to increasing load conditions is proposed, relating this parameter to the time applying the load and, hence, to the Dwell. Crack propagation resulting from creep may also take place under changing load conditions throughout a fatigue cycle, especially when the rate of loading (or unloading) is sufficiently slow to allow creep deformation near the tip of the crack.

The relationship between creep fracture parameters and high temperature is crucial in understanding the behaviour of materials under high-temperature and time-dependent loading conditions. Creep fracture parameters such as *C**, *C(t)*, and *Ct* are used to characterise the rate of crack growth in these environments [49]. The rate of creep strain is highly dependent on temperature, stress, and material composition. Creep deformation is a thermally activated phenomenon, and its rate is governed by an Arrhenius factor, making the temperature dependence a critical factor [49]. In the context of creep–fatigue loading, these parameters play a crucial role in understanding crack growth behaviour, as creep and fatigue effects are the primary mechanisms for failure in such loading regimes [50]. The Ct parameter, in particular, uniquely characterises the rate of expansion of the creep zone size and is essential for predicting crack growth in high-temperature materials such as intermetallics, Ni-based alloys, and high-temperature ceramics [50]. The J-Integral is based on the needed energy calculation in the plastic zone around the crack tip and crack tip stress field; this happens when time is fixed and no transition occurs in steady mode. In order to take the transient creep condition into consideration, *C(t)* was established. Time-dependent conditions make it that both methods cannot be presented in parallel [51].

Tong et al. [52] further elaborate on the significance of Ratchetting Strain as a damage parameter in their subsequent work. They establish the Ratchetting Strain as a driving force for fatigue crack growth. For this purpose, several models, both independent and time-dependent, are developed using elasto–plastic, visco–plastic, and crystal–plastic models [53]. The authors argue that Ratcheting stress always occurs near the crack tip, leading to the progressive accumulation of stress strain normal to the crack growth plane. It seems plausible that this tensile strain, or Ratcheting strain, may be responsible for the material separation leading to crack growth. As mentioned above, the authors introduce several different material models. Subsequently, finite element analyses of the elasto–plastic, visco–plastic and crystal–plastic models are performed, and the results are shown. The correlation of the results obtained with the application of the models with experimental results is presented [53], showing a special agreement when used for the cumulative plastic deformation model. The results obtained for the analytical and numerical models also correlate well. Further experimental results supporting this theory have been published [54].

Complementing this body of work, other research [55] offers a unique perspective by considering crack growth as a process of successive restarts, akin to the initiation stage of a crack. A crack growth model based on accumulated damage was presented in a similar way to a fatigue analysis of stationary notches. Crack growth is considered as the process of successive cracks restarting, as if it was always in its initiation stage. The driving force of crack growth is based on the local stresses and strains around the crack tip, using the Smith–Watson–Topper (SWT) fatigue damage parameter *D = σ_max_ − Δε*/*2*. The driving force of crack growth can be expressed as a function of two parameters: the residual cyclic plastic stresses *σ_res_ (χ)* and the residual intensity factor *K_r_*, which are expressed in different equations. The study shows the correlation of the experimental results and those obtained with the model, validating the use of the model which is based on modelling the crack as a succession of blunt notches of radius *r_p_*, and not as a sharp crack, therefore with realistic stresses and strains around a rounded crack tip.

A different approach is seen in [56] where the focus is on identifying decisive parameters in fatigue crack growth based on accumulated plastic deformation. In this work [56], a numerical study based on the accumulated plastic deformation is presented to determine which parameters are decisive in fatigue crack growth. Firstly, the need to stabilise the crack growth with initial cycles where the da/dN is not stable is established. Secondly, it is compared with experimental results by defining the percentage of accumulated plastic deformation as the driving parameter of crack growth.

Finally, another paper [57] utilises the DIC technique to offer a unique perspective in determining the displacement field around the fatigue crack tip. In detail, the DIC technique was used to determine the displacement field around the fatigue crack tip for a constant value of the stress intensity factor (SIF). The data obtained by DIC was used in the evaluation of the T-strain and to observe its influence on the FCGR. As an original contribution of the study, the higher order terms of the Williams Expansion (WE) were determined. The displacements of a set of points outside the plastic zone were selected for the application of the over-deterministic method (ODM) to obtain several initial WE terms. A numerical model of boundary conditions was also made. The calculated experimental T-stress values showed good agreement with finite element analysis and the literature. It was also shown that the level of restraint influences the fatigue crack propagation rate—the higher the T-stress, the lower the growth.

It is evident that the explored models, while distinct in their theoretical underpinnings, converge in their emphasis on the stress and/or strain history’s role in fatigue crack growth. Beginning with Shih’s [43] foundational work that intricately links the J-integral and CTOD, this section traverses through various models, each underscoring different aspects of crack propagation. The intricate elasto–plastic stress–strain history analysis by Noroozi et al. [39], the extended application of the PICC model by de Matos and Nowell [44], and Chikh et al.’s [40] investigation into the plastic zone’s influence on crack growth rates collectively illustrate the complexity and multifaceted nature of crack propagation studies. Further, the models proposed by Tong et al. [45,46,52] and the novel applications of the DIC technique [57] highlight innovative approaches to understanding the dynamic interactions at the crack tip.

## 3. Damage Theories Based on Crack Growth Concepts—CTOD, Plastic Zone Size, r_p_, ΔK

Following the review, a spectrum of damage theories based on crack growth concepts is presented, emphasizing the interplay between plastic zone size, CTOD, and SIF. Each study contributes to a more integrated understanding of fatigue crack growth, weaving together different theoretical approaches with experimental insights.

In one of the first works [58] to successfully use plastic parameters to model the FCGR, the size of the plastic zone r_pc_ is used as a key parameter. Starting from the equation of r_pc_ based on the Dugdale model, and applying plane strain conditions, Equation (3) is obtained for r_pc_, and with plane stress conditions, Equation (4) is obtained. That is, a specimen of zero thickness would be ideal for plane strain and one of infinite thickness for plane strain conditions. The FCGR is expressed as da/dN = C_e_(ΔK_eff_)m_c_, where C_e_ and m_c_ are material constants. Figure 4a,b show the correlation of the experimental FCGR with that obtained in the model for different specimen thicknesses, as a function of ΔK and ΔK_eff_. Figure 4c,d show the FCGR correlations assuming plane strain and plane stress respectively; it is observed that plane strain conditions favour the FCGR, so it increases with specimen thickness. Following the emphasis on the plastic zone size in [58], the study in [22] extends the discussion by measuring crack length and fatigue crack closure to understand the variation of crack growth rate with SIF under different loading conditions.
(3)$$r_{pc}=\frac{\pi }{18}{\left(\frac{∆K}{{2\sigma }_{y}}\right)}^{2}\left[\frac{1}{2}+\frac{2}{3}\sqrt{\frac{2}{3}}\frac{\sigma_{app}}{\sigma_{y}}\right]$$
(4)$$r_{pc}=\frac{\pi }{8}{\left(\frac{∆K}{{2\sigma }_{y}}\right)}^{2}\left[\frac{1}{2}\frac{\sigma_{app}}{\sigma_{y}}\sqrt{1-\frac{3}{4}{\left(\frac{\sigma_{app}}{\sigma_{y}}\right)}^{2}}\right]$$

The objectives of [22] were to measure crack length and fatigue crack closure to analyse the variation of crack growth rate with *K_max_* or *ΔK_eff_* for different values of *R* (stress ratio) and *B* (specimen thickness), to obtain the *da*/*dN − ΔK_eff_* curves, and to propose an empirical model that correlates a new normalised loading parameter *U* with *R*, *B*, and *ΔK*. The results were obtained for constant load amplitude in tension with three stress ratios: *R* = 0, 0.2, and 0.4, and three specimen thicknesses: *B* = 6, 12, and 24 mm. To measure the crack opening values, gauges were used, and with them *ΔK_eff_*_._ could be calculated. The authors started from a set of equations (in Equations (1)–(3) in the original publication) to relate, in the first instance, *U*, *K_op_*, and *ΔK_eff_*. In view of the experimental results (shown in Figure 4, Figure 5 and Figure 6 in the original publication), an experimental relation between *da*/*dN* and *ΔK_eff_* and a readjustment of U was proposed. As a final model, the relation between *da*/*dN* and *ΔK_eff_* in Equation (5) remains. An important conclusion obtained in the work is that, despite some scatter in obtaining the *da*/*dN − ΔK_eff_* curves, the results of the stress ratios and thicknesses were presented using a simple curve for two parameters of the crack growth rate relationship applied to CK45 steel. A model of *U* as a function of *R*, *B*, and *ΔK* was proposed based on the experimental data of the crack closure. This model is limited to 0 < *R* < 0.4 and 6 < *B* < 24 mm.
(5)$$da/dN=9\cdot 10^{-9}{∆K}_{eff}^{2.95}$$

Building on the understanding of crack closure and stress intensity factors from [22], at the end of the last century [20], Donald and Paris introduced the adjusted compliance method (ACM) and the crack wake influence (CWI), further refining the predictive models for FCGR. In this research, a method for calculating the *ΔK_eff_* was established based on the ACM. Subsequently, the CWI was used to understand the intrinsic limitations of the ACM, and by applying adjustments to account for the CWI and its mechanisms, develop an improved model. Starting from the expression of *ΔK_eff_ = K − K_maxop_*_,_ corrections were applied to arrive at Equation (6) where the parameters exhibit a high degree of similarity (*ΔK_eff_, K_max_* and *K_op_*), but with different nuances.
(6)$${\Delta K}_{eff}={\Delta K}_{op}+\left({\Delta K}_{max}-{\Delta K}_{op}\right)\cdot \left(1-\frac{2}{\pi }\right)$$
(7)$${\Delta K}_{eff}={\Delta K}_{app}\cdot \left(1-\frac{V_{cwi}}{V\cdot IFR}\right)$$

The application of ACR leads to the expression *ΔK_eff_ = ACR − ΔK_app_*_,_ where *ACR* is a coefficient. The CWI method was modelled by subtracting *K_cwi_* from *K* to obtain *K_eff_—K_cwi_* is related to *V_cwi_*, which is the midline shift in a series of equations—to obtain the final expression of *K_eff_* in Equation (7). Two further modifications were proposed based on the adjusted compliance ratio method and the adjusted compliance ratio/opening load blend method. From this point on, the study focuses on representing the FCGR versus the *ΔK*, both experimental and those obtained in the models, an example of which is shown in Figure 5. The FCGR is the *da*/*dN*, expressed in *mm*/*Cycle*, and the *ΔK*, the diverse intensity factors proposed in MPa(m)^0.5^. In total, seven different methods of estimating the effective intensity factor were compared. The ACR method is easy to apply but does not reflect crack closure well. The CWI is mathematically more complex to apply, limiting its applicability. A second version, ACRn2, improves the problems of ACR without adding mathematical difficulty, and finally, the empirical approach for AOP (simply an empirical combination of the ACR and OP methods) makes it a simpler model.

The evolution of the partial crack closure model by Donald and Paris [20] is further advanced by [24], who proposed an improvement to integrate the effects of the R-ratio on crack growth, thereby addressing a crucial aspect that had been systematically neglected. The researcher [24] proposed an improvement to the partial crack closure model proposed by Donald and Paris. The improvement consists of a transition function that evolves from the Donald and Paris partial closure model [20,59] to the model in the region close to the Elber growth threshold [60]. Elber proposed that *ΔK_eff_ = K − K_maxop_*, establishing an effective intensity factor that, in turn, depends on the intensity factor at crack opening. The model proposed in this paper is developed (in Equations (3)–(10) in the original publication), resulting in Equation (8) for *ΔK_effM_*, which adequately correlates the effects of R-ratio on crack growth at low and intermediate load levels in aluminium alloys. Until the publication of this research, the effect of the R-ratio had been systematically neglected. It is important to emphasize that the *R* at the crack tip does not necessarily coincide with that applied to the specimen. It is noteworthy that the key parameter in this model, *ΔK_effM_*, is related to the crack opening profile, which obviously depends on the plastic deformation at the tip. Experimental tests show that the proposed model shows improvements to the model in the Elber growth threshold region.
(8)$${\Delta K}_{effM}=K_{max}-K_{op}\left[1+\left(\frac{2}{\pi -1}\right)e^{-\left(\frac{K_{max}}{K_{maxTM}}-1\right)}\right]$$

Complementing this approach, Zhang et al. [61] introduced a model based on dual parameters, *da*/*dN* and *da*/*dS*, to define the crack propagation velocity, thus broadening the scope of the parameters considered in crack growth modelling. This approach defines the crack propagation velocity with respect to the applied stress at any time of the cycle. Experimental methods were used to predict crack propagation behaviour. The mentioned parameters are related and, following mathematical development, Equation (9) is reached, where *da*/*dN* is expressed as a function of *ΔK*, *α*, and *β*, where *α* and *β* can be estimated from the results of two constant stress amplitude fatigue tests. The correlations of the experimental results with those proposed by the models are shown in Figure 6.
(9)$$da/dN={C\Delta K}^{2\left(\beta +\alpha +1\right)}\left[\frac{1-R^{2\left(\alpha +1\right)}}{{\left(1-R\right)}^{2\left(\alpha +1\right)}}\right]$$

**Figure 6 materials-16-07603-f006:**
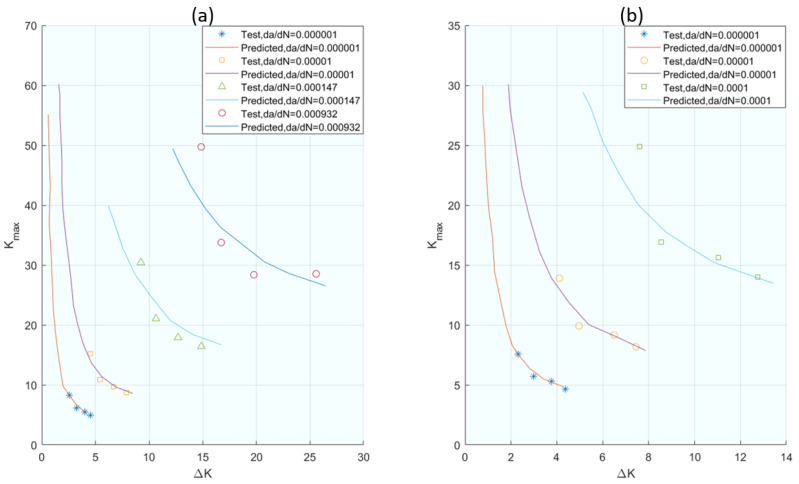
The comparison between the predicted K_max_ vs. ΔK curve and the experimentally obtained data for (**a**) 2324 aluminium alloy and (**b**) 6013 aluminium alloy. From [61].

Further contributing to this comprehensive analysis, Lopez-Crespo et al. [62] undertook an analytical–experimental study focusing on the impact of crack closure and plasticity at the crack tip, thereby offering insights into Mode I and II intensity factors. Through experimental laboratory work, they obtained the displacement field at the crack tip and its environment with DIC techniques on two types of centre-cracked plate (CCP) and wedge opening load (WOL) specimens that had been previously initiated with a crack itself, not a notch. The analytical development is based on formulating the displacement field on the specimen surface as two-dimensional, *u* and *v* being the horizontal and vertical displacements, respectively, and expressing them as a complex number *z = x + iy*. In turn, these displacements can also be obtained from two analytical Muskhelishvili functions *φ = φ(z) ψ = ψ(z)* (Equation (10)). These expressions can be derived by obtaining expressions as a function of *u*, *v*, and the coefficients (*C, D, E, F, α,* and *β*). Applying basic tensor algebra, this can be expressed in a matrix form *Aχ = b* where *b* is the vector that depends on the displacements *u* and *v*. From there, it is straightforward to calculate the coefficients (*C, D, E, F, α,* and *β*) of the matrix *A* and, back in Equation (10), calculate the analytical functions *φ = φ(z)* and *ψ = ψ(z)*. Ultimately, the complex stress intensity factor *K = K_I_ – iK_II_* can be calculated with the above data using Equation (11). Figure 7 plots the intensity factor *K*, the effective intensity factor *K_eff_*, and the real and imaginary components of the intensity factors *K_I_* and *K_II_*. A difference in values obtained with the model and those obtained experimentally exists as the model lacks the crack closure effect, while experimentally there were clear indications of its existence and possible effects.
(10)$${z=\omega \left(\zeta \right)=R\left(\zeta +\frac{\zeta }{m}\right);\varphi \left(\zeta \right)=\sum_{-\infty }^{+\infty }a_{k}\zeta^{k};\psi \left(\zeta \right)=\sum_{-\infty }^{+\infty }b_{k}\zeta^{k};}_{}$$
(11)$$K=2\sqrt{2\pi }\underset{\zeta \to 1}{\lim}\varphi^{\prime }\left(\zeta \right)\frac{\sqrt{\omega \left(\zeta \right)-\omega \left(1\right)}}{\omega^{\prime }\left(\zeta \right)}$$

On a similar note, Zhang [63] developed a fatigue crack propagation model emphasising the effect of compressive stress, thereby adding another dimension to the understanding of fatigue crack growth under tension–compression loading. The stress near the crack tip, the displacement, and the size of the plastic zone were obtained from a kinematically hardened material. The results showed that the above three parameters continue to change with the change of applied compressive force. Experimental methods were used to predict the crack propagation behaviour. The study was based on analyses made by Silva [64,65], who found that materials exhibiting strong cyclic hardening and a high Bauschinger effect (such as Al 2024-T351, ck45, 0.4% mild steel) were strongly affected by the applied compressive load, while materials not exhibiting cyclic hardening (such as Ti6Al4V and Al7175) were relatively insensitive to the applied compressive load. Silva also concluded that models based on the concept of fatigue crack closure are not suitable, while models based on materials with cyclic–plastic properties are suitable for describing fatigue crack propagation behaviour under tension–compression loading. Based on this, Zhang et al. developed a model where they identify the internal and external parameters that control the fatigue crack growth behaviour and describe the fatigue crack propagation rate under tension–compression loading. The paper concludes that, when using a kinematically stiffened material subjected to tension–compression loading, the compression part of the applied cyclic load has a significant effect on the crack tip stress, displacement, and plastic deformation field, and that the maximum applied stress intensity factor *K_max_* and *σ_max-comp_* (maximum applied compressive stress) are the two external loading parameters that determine the above three variables. As an inner parameter, *r_pc_*, the reverse plastic zone size is appropriate to correlate the fatigue crack propagation rate; see also Figure 8. Finally, he proposes an equation for the fatigue crack growth rate under tension–compression loading, with special emphasis that it has nothing to do with the concept of crack closure, but with plasticity at the crack tip and based on the plastic damage theory [61].

Additionally, Pokluda [66] presents a discrete model for small plastic deformations, offering a unique perspective on induced crack closure components, such as plasticity and roughness, and their influence on crack growth. The researcher [66] presents a discrete model for small plastic deformations under plane strain conditions that allows direct evaluation of the magnitude of induced crack closure components, such as PICC and roughness-induced crack closure (RICC). It also determines the influence of the microstructure on the induced roughness. The advantage of the dislocation-based method over those based on continuous mechanisms is its “physical transparency”, which allows the evaluation of both induced plasticity and roughness at the same time. Based on measurements taken on an array symmetrical to the crack, experimental measurements are taken, and plane strain is allowed for the interpretation of the measurements. In this case, the key is to know the stress state on the crack flanks (position of the array); this stress state is the cause of the PICC and RICC. A model is proposed under certain hypotheses for the calculation of the PICC and RICC components, *ΔK, K_max_, K_eff_, K_cl, rs,_ K_cl, rl_,* and *K_cl, p_*_._ The subscript *cl* stands for closure, *rs* for range—short, *rl* for range—long, and *p* for plasticity. Once these characteristic parameters of RICC and PICC are calculated, they can be related to crack growth as deemed best. Finally, *ΔK_eff_ = K_max_ − K_cl_; K_max_ = ΔK*/*(1 − R)*; the expression is given in Equation (12).
(12)$${\Delta K}_{eff}=\left(1-C_{\eta }\sqrt{R_{A}^{2}-1}-\frac{3\eta (R_{A}-1)}{2\sqrt{6}+6(R_{A}-1)}-2C\right)\frac{\Delta K}{1-R}$$

To talk about numerical models of crack growth and their relation to plastic phenomena in the crack tip environment is to talk about Fernando Antunes. Building on the discrete approach by Pokluda [66], Antunes [67] presents an elasto–plastic numerical model, which provides a different perspective by excluding the contact between the crack flanks, thereby focusing on the relationship between *CTOD*, plastic phenomena at the crack tip, and crack growth rate. The work [67] presents an elasto–plastic numerical model where the contact between the crack flanks is obviated. In all simulations, the *CTOD* and *CTOD_P_* were measured. A relationship between *da*/*dN* and *CTOD_p_* was established for the AA6082-T6 aluminium alloy, which is a quantifiable relationship between plasticity and crack growth rate. In addition, the relationship between *CTOD_P_* and other parameters of interest such as ΔK was quantified. When modelling without contact (visible in Figure 4), both plastic and elastic CTODs are higher, i.e., without contact, without the effect of crack closure, the effect of loading is higher. Without contact, without crack closure, the crack propagation only depends on ΔK. The *CTOD_p_* is plotted with respect to the deformation, linear relationship, and with respect to the energy, characterised by a quadratic relationship. In [26], the main objective was to verify the effectiveness of the crack closure concept. Therefore, the crack flank contact was modelled and the following non-linear parameters in the crack tip environment were calculated: cyclic plastic strain range; both elastic and plastic CTODs; size of the plastic inverted zone, *r_pc_* in Figure 8; and the plastic energy dissipated per cycle. Direct relationships of these four parameters with the crack growth rate and *ΔK* are shown throughout the paper, which can be said to form a plasticity-based growth model. The paper also demonstrates the effect of crack closure on the linear parameters, as well as the effect of mesh size on the linear parameters. In another paper [68], the same numerical finite element model is essentially used as in the two previous works, which will be now described briefly: the numerical modelling is assumed to be elasto–plastic with: 1-Isotropy and 2-the Von Mises criterion is followed by the Voce isotropic hardening law [69] and the Lemaitre–Chaboche’s kinematic law of hardening [47]. The numerical model was implemented with DD3IMP in-house code. Only 1/8 of the specimen is modelled by symmetry with the consequent computational savings. The main result of the work is the elucidation of the relationship between the *COTD_P_* with *da*/*dN* for 7050-T6 aluminium. Specifically, there is a linear relationship, with multiplication by 0.5245. Following the basis of the numerical modelling of the previous publications, in [70], the relationship between plasticity and crack growth is further explored and the relationship between *CTOD_P_* and *da*/*dN* for aluminium 2050-T8 is also obtained. As a novelty in this work, the effect of the number of load cycles that elapse until another node is released from the crack tip with the consequent crack growth is studied. The higher the number of load cycles between node openings, the higher the *CTOD_P_*. The da/dN is plotted against the numerical *CTOD_P_*, and the strong influence of the numerical parameters can be seen. In plane strain, the higher the number of cycles, the lower the *CTOD_P_*, unlike in plane deformation. In [67], the methodology for the calculation of the *CTOD_P_* in numerical tests is established in detail, as well as the parameters that will affect this calculation, such as the crack propagation distance, the size of the crack, the distance of the points with respect to the edge, the size of the elements, and the number of previous cycles. The mesh size and the software used are those mentioned earlier in Antunes’ work. The distance of the first released node after the crack edge is the most sensitive parameter together with the mesh size.

Continuing the theme of elasto–plastic analysis, Kawabata [35] proposed a method for *CTOD* calculation that incorporates the impact of strain hardening at the crack tip, further enhancing the understanding of plastic effects on crack growth. This method for *CTOD* calculation [35] was established on the crack tip edge loss due to strain hardening. The researcher introduces a new factor f to account for the plastic factor. This f-factor is a function of the plastic radius and the thickness of the specimen. The result of the correction has been verified numerically and experimentally. Starting from the proposed and revised formulae [71,72,73,74] for the *CTOD* calculation, they concluded that the *CTOD* calculation was overestimated due to the presence of the plastic deformation at the crack tip, which was expected with respect to the edge loss. The *CTOD* is the sum of an elastic and a plastic term; the elastic term is calculated based on the intensity factor *K*, and the plastic term based on the Hinge plastic model. For the calculation of the *CTOD* there are several parameters to adjust: in the plastic part, *m(YR)* is calibrated with Equation (13); the factor *f (YR, B)* of the plastic part is calibrated in Equation (14), which gives as a final result the calculation of the *CTOD* by Equation (15). The goodness of the model has been checked numerically and experimentally, and the correlation is shown in Figure 9.
(13)$$m\left(YR\right)=4.9-3.5YR$$
(14)$$f\left(YR,B\right)=f\left(B\right)f\left(YR\right)=\left[0.8+0.2e^{\left\{-0.019\left(B-25\right)\right\}}\right]\left[-1.4{\left(YR\right)}^{2}+2.8YR-0.35\right]$$
(15)$$\delta_{jwes}=\delta_{el}-\delta_{pl}=\frac{K^{2}\left(1-\nu^{2}\right)}{m\left(YR\right)E\sigma_{YS}}+f(YR,B)\frac{r_{p}\left(W-a_{0}\right)}{r_{p}\left(W-a_{0}\right)+a_{0}}V_{p}$$

Further expanding on the theme of *CTOD*, Vasco-Olmo et al. [75] conducted an experimental study to evaluate the capability of *CTOD* in characterising fatigue crack growth, thereby exploring the practical applications of the theories discussed in previous works. The authors [75] conducted an experimental study of *CTOD* to evaluate the ability of this parameter to characterise fatigue crack growth. A methodology was developed to measure and analyse *CTOD* from experimental data. Displacements were measured by implementing DIC in the fatigue crack growth; in this way the *CTOD* can be determined. Fatigue tests were performed with *R_s_* of 0.1 and 0.6 on compact tension (CT) specimens made of commercial pure titanium. A sensitivity analysis was performed to explore the effect of the selected position behind the crack tip on the *CTOD* measurement. Analysis of a full load cycle allowed the elastic and plastic components of the *CTOD* to be identified. For the plastic, *CTOD* was found to be directly related to the plastic deformation at the crack. Furthermore, a linear relationship between *da*/*dN* and plastic *CTOD* was observed in both tests (Figure 10). The results show that *CTOD* can be used as a viable alternative method to *ΔK* in characterising fatigue crack propagation because the parameter considers the fatigue threshold. This work aims to contribute to a better understanding of the different mechanisms driving fatigue crack growth and the direction of fatigue crack growth, a controversy associated with plasticity-induced fatigue crack closure.

Thermal conditions as a factor in CTOD calculation is presented in [76]. The study links CTOD values with fatigue crack growth across different temperatures. The fracture toughness tests (documented in Table 4 in the original publication) reveal higher CTOD values for both base metal and weldments at cryogenic temperatures compared to room temperature, consistent with previous research. These findings suggest that Al 5083-O demonstrates increased resistance to crack propagation at lower temperatures, as indicated by the CTOD improvement ratio. This ratio, reflecting the comparison of CTOD values between cryogenic and room temperatures, underscores the material’s enhanced fracture toughness under varying thermal conditions.

Lastly, Medhi et al. [77] integrate DIC with analytical solutions to monitor structural health, focusing on the stress intensity factors under cyclic loading, thus connecting the theoretical models with practical structural health monitoring applications. This methodology for structural health monitoring is based on a combination of DIC and an analytical elastic solution. Experimentally (DIC), the full displacement field around the crack tip in a CT specimen made of 2024-T351 aluminium alloy under cyclic loading at different load levels is obtained. Then, an over deterministic multipoint method is used to calculate the SIF in Modes I and II (*K_I_* and *K_II_* in Equation (5) in the original publication). To apply the method, it is necessary to adjust the experimental displacement field obtained with the series described by William [78]. The parameter was determined to facilitate the difference between the SIF result of the (nominal) model and the experimental one. It was observed that, with increasing amplitude in cyclic loading, the difference between nominal and experimental K increased. This may be because of crack tip plasticity, which was not considered in the nominal evaluations. To account for plasticity, Irwin’s approximation was used in the analytical model. The plastic term is added to the crack length, *a_corr_ = a + r_y_*, where *r_y_* is the plastic radius given by the Irwin approximation, $r_{y}=1/2\pi \cdot {\left(\frac{K_{I}}{\sigma_{ys}}\right)}^{2}$. The results showed a better agreement in the evolution of K under cyclic conditions, increasing the load up to the sharp fracture of the specimen.

This section intricately navigates through diverse damage theories centred on crack growth, weaving together the significance of plastic zone size, CTOD, and stress intensity factors. Commencing with the pioneering use of plastic parameters by [58], the discussion evolves, with [22] exploring crack length and closure. This narrative progresses with [20] refining FCGR models through the ACM and CWI methods. The evolution continues with [24] enhancing crack closure models by integrating R-ratio effects, and Zhang et al. [61] expanding the parameters in crack growth modelling. Complementary studies by Lopez-Crespo et al. [62], Zhang [63], Pokluda [66], and Antunes [67] delve into the nuances of crack closure, plasticity, and microstructural influences, while Kawabata [35] and Vasco-Olmo et al. [75] focus on CTOD methodologies. Lastly, Medhi et al. [77] bridge these theoretical models with practical structural health monitoring applications. Collectively, these studies form a cohesive narrative, significantly enriching the understanding of fatigue crack propagation and the importance of CTOD and crack tip plasticity that are demonstrated through experimental work [67,75,77].

## 4. Energy-Based Theories of Damage

Expanding on the concept of energy accumulation and/or dissipation as a fundamental factor in fatigue crack growth, this section delves into various models that utilise energy-based theories to elucidate the damage process in ductile solids. Each model, while grounded in the same theoretical domain, offers unique perspectives and methodologies, contributing to a broader understanding of crack propagation dynamics.

In one of the earliest works on this topic, Klingbeil [79] proposed a new theory of fatigue crack growth in ductile solids based on the total plastic energy dissipation per cycle ahead of the crack. The fundamental hypothesis of the theory proposes a unified criterion for crack growth under monotonic conditions and fatigue loading, so that the fatigue crack growth rate is given explicitly in terms of the total plastic dissipation per cycle, the monotonic plane strain or plane stress, and the fracture toughness of the material. Here, the total plastic dissipation per cycle was obtained by 2D elastic–plastic finite element analysis of a stationary crack with constant amplitude loading of Mode I CT specimens. Initially, the aim was to exclusively relate the fatigue crack growth to the total plastic energy dissipated, emulating the concept of the J-integral as a crack growth parameter. For the calculation of the dissipated energy, a stationary numerical model was proposed, in which the possibility of crack closure is not given. The FEM was implemented with ABAQUS, which, under small elasto–plastic deformations and Von-Mises criterion, reduced integration to abolish shear stiffness and minimum element sizes less than 0.05 µm. Figure 11a shows how the parameters affect the energy; the dissipated energy is directly calculated by ABAQUS as shown in Figure 11b. It was obtained as a result that the dissipated energy decreased with hardening. As a result, Equation (16) fits a crack growth model as a function of dissipated energy.
(16)$$\frac{da}{dN}*=\frac{\frac{da}{dN}}{\left[{\left(\frac{K_{Ic}}{\sigma_{y}}\right)}^{2}\frac{dW}{dN}*\right]}\;and\;{∆K}^{*}=∆K/K_{Ic}$$

Transitioning from the specific focus on plastic dissipation in [79], Pommier and Risbert’s study [38] introduces a more holistic approach by considering a supplementary state variable for the crack, enhancing the understanding of the energy dynamics in fatigue crack growth. Pommier and Risbert [38] presented a paper where the main objective was to propose a set of equations time derived for fatigue crack growth in order to avoid any deconstruction of the cycle. The model is based on the thermodynamics of dissipative processes. Its main originality lies in the introduction of a supplementary state variable for the crack, which allows the crack state to be described continuously throughout any complex loading sequence. The crack state is fully characterised on a global scale by its length a, its plastic blunting ρ, and its elastic opening *∆CTOD*. The plastic blunting is calculated by integrating the difference of the elasto–plastic and elastic displacement at the crack tip along d, and dividing the result by d. The model starts from an energy balance of the crack where *D* is the dissipated energy, which is formulated in Equation (17). Together with the Clausius Duhem inequality, they are combined in the expression of *da*/*dN* in Equation (18). The model finally consists of two laws: a crack propagation law, which is a relation between dρ/dt and *da*/*dt* and observes the inequality derived from the Clausius Duhem inequality, and an elasto–plastic constitutive behaviour of the cracked structure, which gives dρ/dt versus load and is derived from the energy balance equation. By solving the set of equations with the appropriate boundary conditions, the evolution of the different crack parameters with time is obtained. The model was implemented and tested. It successfully reproduces the main features of fatigue crack growth as reported in the literature, such as the Paris law, the stress ratio effect, and the overload retardation effect.
(17)$$D=\left(\phi_{a}^{eff}\frac{da}{dt}{+\phi }_{\rho }^{eff*}\frac{d\rho }{dt}\right)+\left(\phi_{a}^{\chi }\frac{da}{dt}{+\phi }_{\rho }^{\chi *}\frac{d\rho }{dt}\right)$$
(18)$$\frac{da}{dN}=\int_{t(N)}^{t(N+1)}\frac{da}{dt}dt=\frac{\alpha }{2}\left(\rho_{max}-\rho_{min}\right)\approx \frac{\alpha }{2}\frac{∆CTOD}{2}$$

Further advancing the energy-based perspective, ref. [80] shifts focus to the cumulative change in the cyclic strain energy, offering a complementary view on how energy parameters drive crack growth under various loading conditions. The aim of [80] was to demonstrate that fatigue crack growth under elasto–plastic conditions can be perfectly correlated using the concept of physically finding a driving force parameter. In this work, the selected parameter is the cumulative change in the cyclic strain energy of the net section, thereby necessitating a physical approach independent of geometry and loading conditions, and establishing a parameter as the driving force of the crack, which is calculated analytically from the elasto–plastic behaviour of the material and the relative sizes of the crack sections in their corresponding planes. The author demonstrates and calculates the stresses (σ), strains (ε), elastic and plastic energies (U), and idem with the energies of the net cross-sections (U_L_, in Equations (1)–(14) in the original publication). The applied loads are cyclic in both stress and strain control. The author presents an approach based on the treatment of elasto–plastic deformations in the net section, from the material resistance point of view. The physical driving parameters of the crack are calculated as the difference of the energy of the net section and the energy before the crack appeared. ΔC_pσ_, Equation (19), for stress-controlled fatigue and ΔC_pε_, Equation (20), for strain-controlled fatigue. Both parameters are expressed in terms of material hardening and deformation parameters (k, n), crack size (a), and specimen width (W). It achieves excellent correlations between the crack growth ratios as a function of the strain energy parameters in the net section as seen in Figure 12. Furthermore, the correlations were obtained without any consideration of the concept of crack closure. As a conclusion, it was also added that this approach can be used in many inelastic deformation situations, such as creep crack growth and crack growth in viscoelastic materials.
(19)$$\Delta C_{p\sigma }^{sL}=\frac{n\left[\sigma_{\frac{n+1}{n}}^{\left(\infty \right)}-\sigma_{\frac{n+1}{n}}^{\left(o\right)}\right]}{K^{\frac{1}{n}}\left(n+1\right)}\left[{\left\{\frac{W}{W-a}\right\}}^{\frac{1}{n}}-1\right]$$
(20)$$\Delta C_{p\in }^{eL}=\frac{K\left[\in_{\left(n+1\right)}^{p\infty }-\in_{\left(n+1\right)}^{o}\right]}{n+1}\left[\frac{a}{\left(W-a\right)}\right]$$

Finally, ref. [81] propels the energy-based discussion forward by proposing a comprehensive energy equation for fatigue crack growth. This approach seeks to quantify the various energy contributions per cycle, thus providing a more detailed understanding of the energetic interactions during crack propagation. An energy equation for fatigue crack growth is therefore proposed [81]. It equates the total external work per cycle (*dW*/*dN*) to the sum of the plastic dissipation (*dU_pl_*/*dN*), the increase in stored strain energy (*dU_e_*/*dN*) and the energy dissipated in the formation of new crack surface (*dU_a_*/*dN*). Experimental measurements of fatigue crack growth were performed to obtain the relationship between fatigue crack growth rate (*da*/*dN*) and energy variables. The result shows that *dU_pl_*/*dN* and *dU_e_*/*dN* are not directly related to *da*/*dN*. The *dU_a_*/*dN*, whose value cannot be obtained experimentally with sufficient regression, may be the variable directly related to *da*/*dN* within the test range.

As presented in previous sections, temperature is a topic worthy of discussion. Sohail and Roy [82] explore energy-based models for fatigue crack growth by linking the J-integral with entropy and high temperatures. They investigate the applicability of linear elastic fracture mechanics (LEFM) at the atomic level in amorphous materials and quantify the entropic contributions to the J-integral at a nanoscale, particularly under thermal conditions. The study uses molecular dynamics to demonstrate how the atomistic J-integral can be employed to determine cohesive traction–separation laws in polymers, highlighting the significance of temperature and notch size on fracture toughness and strength.

A comprehensive examination of energy-based theories in the context of fatigue crack growth is presented, with each model contributing uniquely to the collective understanding of this complex phenomenon. The works of [38,79,80,81] collectively underscore the pivotal role of energy dissipation and strain energy parameters in delineating crack propagation dynamics. From [79]’s exploration of total plastic energy dissipation to [38]’s innovative introduction of a supplementary state variable for crack characterisation, and onto [80]’s focus on the cumulative change in cyclic strain energy and [81]’s holistic energy equation, these studies reflect a concerted effort to quantify and understand the energetic interactions at play in fatigue crack growth. This cohesive body of work not only reinforces the significance of energy considerations in crack growth studies but also illuminates the relation of various energy forms in influencing crack propagation under diverse loading conditions.

## 5. Hybrid Damage Theories or Parameter Definition

This section explores a range of hybrid damage theories and parameter definitions, emphasising the intricate interaction between microstructural components, energy dissipation, and crack growth dynamics in various materials. Each study, while distinct in its approach, contributes to a deeper understanding of fatigue damage, highlighting the importance of multi-dimensional analysis in fatigue crack growth models.

The work [15] focuses on the high cycle fatigue (HCF) life in cast Al-Mg-Si alloys, which is particularly sensitive to the combination of microstructural components, inclusions, and stress concentrators. Inclusions can range from large-scale shrinkage porosities to trapped oxides introduced during casting. When controlling for shrinkage porosity, the relevant microstructures at nucleation sites are often the larger Si particles within the eutectic regions. In this paper, an HCF model is introduced that recognises multiple scales of inclusions for crack formation. This ambitious work presents a model that addresses the role of constrained microplasticity around detached particles, shrinkage pores in the formation, and the microstructural growth of small fatigue cracks. It is based on cyclic crack tip displacement rather than the linear elastic fracture toughness factor. The model is applied to an aluminium alloy A356-T6. A relationship between *da*/*dN*, *Δσ*, *S_u_*, *A*, and *ΔCTOD* (*S_u_* is the maximum force) is presented. Seemingly uncomplicated, the authors attempt to outline all the possibilities and steps (Equation (1) in the original publication). For example, the number of cycles is expressed as *N_T_ = N_INC_ + N_MSC_ + N_PSC_ + N_LC_*, where *INC* is incubation, *MSC* is propagation of a microstructurally small crack, *PSC* occurs during the transition from *MSC* status to that of a dominant long crack, and *LC* is long crack. These cycle numbers are related to the porosity size using various coefficients. For the MSC/PSC regimes, the cycle numbers are also expressed as a function of the most characteristic parameters of the process according to the authors. The whole process detailed so far is repeated for five types of inclusion models: (a) distributed microporosity and Si particles—no significant pores or oxides; (b) high levels of microporosity—shrinkage or gas pores with maximum diameter *D_p_* < *3DCS* but greater than the maximum Si particle diameter; (c) large pores near the free surface (*D_p_* > *3DCS*); (d) large pores near the free surface (*D_p_* > *3DCS*); (e) large oxide films. The model results are correlated with the experimental A356-T6 plate results (in Figure 4, Figure 5, Figure 6, Figure 7, Figure 8, Figure 9, Figure 10 and Figure 11 of the original publication). The work ends with some interesting graphs (Figure 13 and Figure 14 in the original publication), where the size of the inclusions is related to the crack growth mechanisms, as well as to the fatigue life of the materials. Transitioning from the microstructural emphasis in [15], the subsequent study by Khelil et al. [83] introduces an energetic perspective, illustrating how different approaches can converge to enhance the understanding of fatigue crack growth.

Khelil et al. [83] present an energetic approach to fatigue crack growth (FCG). This approach is based on the numerical determination of the plastic zone by introducing a new form of plastic radius. Experimental results regarding two aluminium alloys of the types 2024-T351 and 7075-T7351 were used to validate the developed numerical model. A good agreement between the two types of results was found. Starting from Equation (21), *ΔW_p_* is formulated as the plastic energy of the hysteresis cycle characteristic of cyclic loads and *U* as the Specific Energy. *U*, Equation (22), is expressed as a function of *da*/*dN* and subdivided into the three phases of the kinetics fatigue failure diagrams (KFFD). On the other hand, taking as valid the value of the angular strain γ = ε(3)^0.5^ near the crack tip, one can express *R_N_ (ϴ)* as the singular dominant term approximating the elasto–plastic boundary. This allows further development of the *ΔW_p_* which in Equation (21) is left as an integral of stress and strain terms alongside *r_p_* (plastic radius), *S_pz_* (plasticised surface), and *Q*, Equation (23) (total energy dissipated).
(21)$$∆W(PZ)=2\left(1-N^{\prime }\right)\frac{1-N^{\prime \prime }}{1+N^{\prime \prime }}{∆\sigma_{0}∆\epsilon_{0}\left(\frac{∆K_{I}}{∆\sigma_{0}}\right)}^{4}\int_{0}^{\frac{\pi }{2}}f_{N}{\left(\theta \right)}^{2}d\theta$$
(22)$$U=\frac{M}{2B\cdot da/dN}$$
(23)Q=∆W(PZ) · B

These expressions are a function of parameters that must be calculated iteratively as indicated in the flow diagram (Figure 7 in the original publication). Figure 13a,c shows the correlations of dissipated energy and ΔK and Figure 13b,d does the same for *da*/*dN* and *Q* with acceptable correlations. Finally (in Equations (35)–(37) in the original publication), the relationship of *da*/*dN* and *ΔK* for the three phases of the KFDD is established.

Building upon the energetic approach of [83], the next study [84] broadens the scope by integrating experimental studies and modelling techniques, thus providing a comprehensive view of the fatigue crack growth rates in aerospace alloys. The work [84], apart from the crack growth model due to dissipated plastic energy, also presents an overview of experimental studies and modelling of fatigue crack growth rates in the aerospace titanium alloy Ti-6Al-4V. Firstly, numerous experimental tests were carried out on CT specimens of the alloy, from which the following was obtained: the FCGR under constant load. The influence of various parameters such as crack length and the effect of overload (retardation) were also evaluated experimentally. The crack opening was measured using DIC with a digital camera equipped with a Questar wide-range telescope; this set-up can record at 0.1 µm pixel quality. The residual SIF (*K_R_*) was determined using X-ray diffraction techniques with a synchrotron, which also served to map all the deformation in the vicinity of the crack tip. The modelling techniques used for FCGR prediction are based on three concepts: (i) Crack advancement is controlled by the damage processes occurring within the fracture zone located ahead of the crack tip. This zone is embedded within the plastic zone of the crack tip, which, in turn, is surrounded by the zone of dominance of the elastic mechanics stress field solution (*K_field_*). (ii) Prediction of crack advance should be possible by knowing the deformation of a small volume around the crack tip, including residual stress and damage accumulation. (iii) Strictly speaking, a distinction must be made between plasticity and damage: although both are dissipative mechanisms, ultimately it is the damage component that determines failure. However, in many metals, the two parameters are related, i.e., it can be assumed that the damage at each point in the material is a function of the plastic deformation. Therefore, FCGR must be correlated with the plastic deformation processes at the crack tip. To have a good growth model, it is essential to choose the parameters well. In this study, the relationship between crack tip blunting ρ and crack length *a* is first obtained. The FCGR is related to the residual intensity factor, the ratio of cyclic loads *R*, and a series of adjustable constants. The adjustment and calculation of the parameters and constants leads to the equation where the FCGR is related to the dissipated plastic energy and a coefficient *β*; this equation implies the notion that a part of the dissipated plastic energy is converted into damage, causing propagation. Finally, the parameters $\rho_{p}$ (crack tip blunting) and *ΔW_p_* (plastic dissipate energy) are chosen as the constituents of the model whose results are compared with the experimental ones shown in Figure 14, offering a more than satisfactory correlation with better results for the blunting parameter.

Further exploring the realm of fatigue crack growth, the research by [85] delves into the complexities of plane strain cracks, challenging some prevailing notions and suggesting new directions for future modelling efforts. The aforementioned paper does not present a model per se, but rather evaluates some parameters as possible future keys in models and discards others. The paper [85] presents the finite deformation elasto–plastic analysis of plane strain cracks subjected to cyclic Mode I loads with constant amplitude at various load ranges and ratios, as well as with overload and underload cycles. The Laird–Smith mechanism of crack growth by cyclic blunting and re-sharpening, which transfers material from the crack tip to the crack flanks, is visualised. In the present model, crack closure has never been detected. Furthermore, the assumed origin of PICC is ruled out. However, the simulated fatigue crack growth (FCG) by blunting and regrinding reproduces the key experimental trends related to the effects of ΔK and single peaks of overload/underload. The calculated curves are kinked despite the absence of PICC, raising doubts about their reliability as a means of crack closure detection and assessment. Thus, the modelling performed manifests ambiguities with respect to PICC as the universal intrinsic factor capable of uniquely controlling FCG. On the other hand, in the absence of PICC, the calculated stress–strain responses near the tip, being the driving forces of fatigue damage and bond-breaking crack advance, manifest affinities with experimental FCG trends without the intervention of PICC. This implies both independent parameters of cyclic crack tip loading under small-scale creep, such as the pair *K_max_* and *K_min_*, or *K_max_* and *ΔK*, or other equivalent, as well as the indispensable variables that drive FCG directly without PICC mediation. One of the major problems of looking at the influence of PICC on crack growth is that under plane strain conditions PICC is not observable. Therefore, numerical modelling appears to be a good tool. The finite element models use a perfect elasto–plastic model with Von-Mises criterion associated with perfect plastic flow. No contact between the flanks was introduced, nor was meshing used. To maintain this criterion, the number of cycles was lowered, and different mesh models with very small element sizes were used. The most remarkable thing according to the authors is that the effect of PICC on FCG seems to be observable, even though PICC has not been modelled, so it is questionable whether PICC affects FCG or not.

Ultimately, the study in [86] evaluates the application of strip yields models, further enriching the discourse on fatigue crack growth under random loading, thereby showcasing the diversity and depth of hybrid damage theories. In this work [86], the application of strip yields models implemented in NASGRO 6.02 (software) to estimate fatigue crack growth under random loading is evaluated. The two different strip yields model options (constant constraint–loss (CCL) and variable constraint–loss (VCL)) are evaluated and compared. The CCL is very similar to Newman’s FASTRAN fatigue crack closure model. In this model, the tensile restraint factor *α* is constant along the plastic zone elements, but its value depends on the stress state, ranging from plane strain to plane stress. This constraint–loss is based on the observation that cracks that initially start with a flat face will eventually grow in a tilted-face manner. Newman proposed that the transition occurs when the size of the cyclic plastic zone (calculated from Δ*K_eff_*) reaches a percentage of the thickness. In the second, VCL, the tensile restraint factor *α* varies along the plastic zone according to a parabolic expression. The constraint decays from its value at the crack tip (*α_tip_*) to a flat stress value of 1.15 at the end in front of the plastic zone. The loss of restraint is also evaluated, but unlike the previous model, the value of *α_tip_* in both strain and plane strain is calculated from the ratio of the size of the plastic zone to the thickness of the specimen. The FCGR is formulated (Equation (24)), where *C*, *n*, *p*, and *q* are parameters that can be adjusted thanks to the NASMAT module. Three combinations of these parameters were considered, and the results compared with data from the NASGRO materials database and the literature. The ability of the models to estimate fatigue life and variability was analysed by comparing the simulated results with experimental fatigue crack growth data under different stationary Gaussian random loading processes on 2024-T351 aluminium CT specimens. The analysis showed that the two models with their three configurations provide good fatigue life predictions, with very similar results. The variability of the results due to the randomness of the loading was also analysed, and in this case the CCL model provides a better estimate than the VCL. Finally, it should be noted that the best correlation with the experimental results was achieved with one of the combinations of constants proposed by the authors. This combination implemented with the NASMAT module improved upon any combination using the NASGRO database.
(24)$$da/dN={{C(\frac{1-f}{1-R}∆K)}^{n}\frac{{\left(1-\frac{{∆K}_{th}}{∆K}\right)}^{p}}{{\left(1-\frac{K_{max}}{K_{c}}\right)}^{q}}\;}^{}$$

The models presented offer a variety of hybrid damage theories and parameter definitions, each contributing a unique lens through which the complex phenomenon of fatigue crack growth can be understood. From the detailed exploration of microstructural influences on high cycle fatigue in Al-Mg-Si alloys [15], to the energetic approaches of Khelil et al. [83] and the comprehensive experimental and modelling techniques applied to aerospace titanium alloys [84], these studies collectively underscore the multifaceted nature of fatigue damage. The critical evaluation of prevailing notions in [85], alongside the implementation of strip yield models in [86], further exemplifies the dynamic connection between theoretical modelling and practical applications. This diverse array of research not only enhances the knowledge in fatigue crack growth across different materials and conditions but also emphasises the necessity of integrating various theoretical and experimental approaches to develop more robust and comprehensive fatigue damage models.

## 6. Theory of Critical Distances (TCD)

This section delves into various applications and adaptations of the theory of critical distances (TCD), a key concept in understanding fatigue crack growth. While each model presented here is based on TCD, they offer unique adaptations or applications of this theory, showcasing the versatility and broad applicability of TCD in different contexts.

In works [87,88], a numerical model based on the theory of critical distances (TCD) is proposed. This theory is not new [7,89], and essentially consists of ensuring that crack propagation will occur when a certain physical quantity, at a given location in front of the crack front, reaches a specific value. In this work, this theory is partially modified to model the crack growth rate, the point method (PM) is adopted, and the physical quantity will be the plastic energy dissipation in front of the crack tip, at the critical point, similar to Figure 8. The numerical model was developed with Abaqus, under plane strain conditions due to the thin thickness of the specimens; a total of 7667 four-node elements with reduced integration were used and the minimum mesh size in the crack was less than 10 µm. Crack closure was modelled with the contact of the fronts; the linear penalty method coupled with an augmented Lagrangian algorithm was implemented to improve the accuracy of the overpressure during closure. The frictionless sliding of the crack plane along the *x*-axis of contact was also allowed. One of the critical points of this work, evidently, is the calculation of the plastic energy dissipated in each cycle for its parametric use as damage, and, therefore, causing the fatigue crack advance. For this purpose, an equation is proposed that essentially integrates the stress over the plastic strain differential. To model the crack propagation, the mesh size must be considered in the order of 2.5–7.5 µm with the actual growth rate being around 0.1 µm/cycle. It quickly follows that many cycles must accumulate in the numerical model before the next node is released. During this accumulation of cycles, there is a stationary regime which becomes transient when the energy accumulates at the critical distance and the crack propagates, releasing the corresponding node. The modelling of crack propagation requires a specific post-processing program. Two are the key input parameters, the *L_CD_*, the critical distance length, and the *E_C_*, the critical plastic energy. The determination of these parameters is of vital importance. For this purpose, the *da*/*dN* vs. Δ*K* curves were determined experimentally. Figure 15 plots the numerical results obtained for the *da*/*dN* vs. ***Δ****K* curves for different values of *L_CD_* and *E_C_*. Finally, the parameters were fitted with the least-square method at *L_CD_* = 17.5 µm and *E_C_* = 0.55 J/mm^3^. Good correlations between the model and the experimental results were observed.

Building upon this foundation, subsequent studies further explore the intricacies of TCD, examining how factors like mesh size and microstructure size, along with variations in loading conditions, influence the outcomes predicted by the TCD-based models. An important step is to study how mesh size and microstructure size affect these outcomes; the latter was investigated because the adjusted critical distance is of the order of the microstructure size, and several authors have previously focused on this detail [90,91]. Subsequently, in another paper [88], the same authors tested the goodness of the method against varying conditions of overload, underload, acceleration, and deceleration of the loading cycles, as well as the moment (crack length) at which this is done. The model responded satisfactorily with respect to the experimental results obtained. The aim of [92] was to predict the fatigue life of 6201-T81 aluminium alloy wires containing geometric discontinuities using the TCD. The equivalent stress was evaluated by the PM and the volume method (VM), based on the maximum principal stress (σ_I_). Regarding the latter method, the relationship between the critical distance and the number of cycles to failure was calibrated using two different methodologies. When life estimations were performed based on the first calibration, the results were considered unsatisfactory. However, when the second calibration method was used, almost all predictions fell within scatter bands with respect to the experimental data used to calibrate the model.

LCF prediction is critical to ensure the structural integrity of an engine. The concept of strain energy gradient was developed [93] and a general procedure established that combines the energy concept with the critical distance theory for life prediction of components with notches in LCF. There is a relationship of the strain energy distributed within the effective damage zone and fatigue crack growth. For specimens with GH4169 and TC4 notches, as well as in a case study of a high-pressure turbine disc, the proposed procedure provides better life correlations than the Fatemi–Socie and Smith–Watson–Topper models.

As a continuous advance in TCD, ref. [94] aimed to compare the predictive capabilities of TCD based on a damage parameter; specifically, four were analysed. The study also relates local strains and stresses to evaluate the fatigue life of notched components subjected to bending–torsion. The fatigue damage parameters tested were defined using well-known stress-based, strain-based, SWT-based, and energy-based relationships (Table 1). Multiaxial cyclic plasticity in the notch zone was modelled with 3D finite elements, using three local stress–strain approaches: Neuber’s rule, the equivalent strain energy density rule (ESED), and the modified rule (ESEDM). Neuber’s rule always led to more conservative results, and the ESEDM rule gave slightly better fatigue life predictions compared to the original ESED rule. As for the fatigue damage parameters, the energy-based models were more accurate.

In essence, the studies in this section demonstrate a comprehensive and multifaceted approach to applying the TCD, reflecting its adaptability and relevance across a spectrum of scenarios and materials. The diverse methodologies, ranging from the nuanced analysis of plastic energy dissipation to the integration of strain energy gradients and various damage parameters, collectively enrich the understanding of TCD and its practical applications in predicting fatigue life and crack growth.

## 7. Conclusions

In concluding this comprehensive review, the role of crack tip plasticity in shaping fatigue crack growth models has been thoroughly explored. This paper seamlessly integrates numerical simulations and analytical methods, meticulously evaluating various approaches to emphasize their merits and constraints. The vital synergy of experimental data with theoretical and simulation-based models is illuminated, providing a detailed perspective on the progression of fatigue crack growth analysis.

In the realm of methodological advancements, a substantial shift towards integrating more sophisticated computational models, particularly finite element modelling, with experimental approaches was observed. This hybrid methodological approach has enhanced the precision of fatigue life predictions, allowing for more accurate simulations of real-world material behaviour under various stress conditions. The application of finite element modelling has been particularly noteworthy in the context of complex loading conditions and the study of microscopic fracture mechanisms.

In these advances, it is of huge interest to pay attention to the work of de Matos and Nowell [44], who proposed modified models for diverse geometry and the possibility of incorporating residual stress fields. The importance of this lies in the ability to model fatigue problems with variables that in most cases cannot be neglected in real engineering applications. Models that are able to reproduce better real-signs are highly valuable. Due to the different approach taken by Mikheevskiy et al. [55], their research is worth mentioning. New approaches and ideas might be the key that is needed in the field to provide real solutions to a problem that, still today, is not yet fully understood.

For simplification purposes, summary tables (Table 2, Table 3 and Table 4) have been drawn up for the most used parameters in the literature. The information shown in the tables consists of the bibliographical reference with its first author and date of publication, as well as a brief description of the contribution with respect to the parameter in question. It also shows the methodology used in the research, or the methodologies in case there are several of them. The authors of this review subjectively suggest how positive (+), very positive (++), or extraordinarily positive (+++) the contribution of the research has been to advancing our knowledge and the relationship that the parameter has with crack growth. This suggestion has an objective basis based on the number of citations of the paper, as well as another aspect of subjective evaluation in which the authors of this review have scored according to their personal experience. Table 2 condenses the information on the CTOD and the CTOD_P_. The latter parameter has been proposed as replacing even the CTOD; the advantages of this are varied, from the unity of the parameter m, in the case of the CTOD, and above all the elasto–plastic nature of the CTOD as opposed to the elastic approach of ΔK. At present, it is a well-established hypothesis that the origin and first stage of the crack has a plastic and microstructural basis, while the second growth phase is of an elastic nature and much more influential on a macro level. Any numerical or analytical approach needs experimental evidence of this. In this direction, the works of Vaso-Olmo et al. [75] and Medhi et al. [77] give insights into the concept of CTOD and plasticity in the crack tip. Both works are able to explain the fatigue crack growth behaviour with an introduction to crack tip plasticity in different ways. It is important to highlight Antunes et al.’s latest studies [67,68,70] for their exceptional contributions to advancing the understanding of the implications of crack tip plasticity in fatigue crack growth predictions based on numerical models with validation with experimental results. In summary, this research successfully established a robust method for the numerical calculation of CTOD and CTOD_P_, emphasising their dependency on key parameters. The study initially focuses on the 7050-T6 aluminium alloy, where cyclic plastic deformation is meticulously determined through experimental tests and subsequently modelled analytically. The development of a 3D numerical model to predict CTOD_P_ marks a significant advancement in this area. Experimental investigations provide a foundational understanding of the relationship between CTOD and FCGR. This is further complemented by numerical predictions of CTOD_P_, which are tailored for varying crack lengths and da/dN values, thereby enhancing the precision and applicability of the model in predicting fatigue crack growth.

The study conducted by Pommier and Risbert [38] deserves recognition for its novel approach as it represents a significant stride in fatigue crack growth modelling by introducing plastic behaviour as an energy-based parameter evolving over time. Their novel approach, rooted in the thermodynamics of dissipative processes, uniquely incorporates a supplementary state variable for the crack. This allows for a continuous and detailed description of the crack state through complex loading sequences, a concept widely acknowledged and supported within the research community. The integration of plastic blunting, an established and well-regarded concept, into this framework further strengthens the model’s applicability. By correlating the thermodynamic process of plastic energy dissipation with the hardening and sharpening of the crack tip growth, the model adeptly captures key aspects of fatigue crack progression, such as the Paris law and overload retardation effects. This work not only advances the understanding of fatigue crack dynamics but also sets a new precedent for incorporating time-dependent plastic behaviour into predictive modelling.

A significant portion of this review delved into the theoretical foundations underpinning the study of fatigue and fracture in materials. The TCD, a pivotal concept in understanding crack propagation and fatigue life, received considerable attention. Branco [94], in the context of assessing the fatigue life of notched components subjected to bending–torsion loading, conducted a comparative analysis to evaluate various local approaches. It was observed that the implementation of ESEDM techniques in 3D finite element models resulted in a higher degree of precision in the prediction of fatigue life.

In Table 3, geometrical parameters that are used to measure some property or definition of the plastic zone are collected, with a suggestion column also added here. Without obtaining a valid model for all situations, there are many investigations that manage to obtain very reliable correlations between the geometric parameters that define the plastic zone and crack growth.

Incorporating hybrid damage theories in this review is essential to explore alternative research avenues beyond the established ones. It is important to mention McDowell et al.’s [15] contribution about the sensitivity to the combination of microstructural components, inclusions, and stress concentrators. Meanwhile, Khelil et al. [83] introduces an energy approach through a numerical model, using of a new form of plastic radius that has been validated with experimental results. Korsunsky et al. [84] presents an energetic approach through the integration of modelling techniques and experimental studies in aerospace applications.

Table 4 presents a summary of the main lines of research that have in common some aspect of plastic energy as a driving parameter in crack growth. It is wise to mention the differences between the terms in Table 4. They can be divided into groups: terms related to plastic strain energy, elastic strain energy, strain energy gradient, total strain energy, and specific energy. The terms related to plastic strain energy are plastic CTOD [67], plastic energy in the point used in PM [87,88], Kujawsky and Ellyin plastic energy [90,91], ΔW_p_ [79,83,84], U_pl_ [81], and Q [83]. All of these consider a threshold of the plastic energy that triggers the crack growth when surpassed. Nevertheless, there are some differences between them; while [79,83,84] refer to the same term ΔW_p_ related to load cycle, other terms like [90,91], U_pl_ [81], and Q [83] are related to crack or fracture cycle release. Quan and Alderliesten [81] also refer to the variation in elastic strain energy in one entire crack propagation cycle, which is included inside the elastic strain energy group. The strain energy gradient is mentioned in [90,91,93], with the difference that Zhu et al. reflects on stress and strain effects. Regarding the group where total strain energy is mentioned, terms like ΔWt [94], U_L_, ΔCpσ and ΔCpε [80], and Ua [81] are included. While ΔWt [94], U_L_, and ΔCpσ [80] are stress-related parameters, ΔCpε [80] is strain-related, and Ua [81] is surface-related. The last term is linked to specific energy, U [83], energy dissipated per unit volume during fatigue crack growth.

The existing problems are based on the absence of a physical basis for most of the proposed models. This makes models based on experimental results representative only if the operational conditions are the same. Something similar, but with a different nature, occurs in finite element models, where boundary conditions, mesh size, analysable parameters, etc., are imposed. FEMs must be validated with experimental results and serve to extrapolate new test conditions; numerical simulations are much more efficient and economical than experimental tests. However, one must not lose sight of the main objective of fatigue research, which is mathematical modelling based on physical phenomena that are common to fatigue cracks. As previously mentioned, in the problem of fracture and fatigue, the microstructural and macrostructural levels, as well as the plastic and/or elastic domain, intersect, representing a handicap for its physical and mathematical modelling.

The difficulty in this case lies in how to measure or estimate experimentally such specific aspects of the energy in such a small space as the plastic zone at the vertex of the crack. Furthermore, analytically it presents an enormous challenge, because, to date, an energetic study with Lagrangian or Hamiltonian mechanics has not been published to date (except for by error or omission), which were developed for an energetic approach to physics [95].

## 8. Future Directions

The field continues to advance through the interplay of theoretical development, methodological innovation, and practical application. Looking into the future, the ongoing challenges and emerging research directions promise to further deepen our understanding of material behaviour. The synergy between experimental research and computational modelling will be crucial in this research field, enabling researchers to predict, understand, and ultimately control material behaviour under the most demanding conditions.

### 8.1. Advancements in Multiscale Approaches

The development of universally applicable models for predicting fatigue life across diverse materials and loading conditions remains a primary challenge in material science. The various theories in the field of fracture and fatigue mechanics have generated a disparity that distances the link between the theory of continuous media (macroscopic approach) and the formulation of particle systems (microscopic approach), hindering the search for a point of convergence. In this sense, the need to explore a rapprochement between the two perspectives is evident, given the potential benefit that could be derived from the synergy between these two apparently disparate approaches.

It is interesting to note how, in a similar context, the expert Pradl achieved an important breakthrough in the field of fluid mechanics by bringing apparently divergent theories closer together.

### 8.2. Challenges in Experimental Measurement and Analytical Modelling

One of the major challenges in the field is the accurate measurement and estimation of energy within the plastic zone at the crack tip. The small scale of this area makes it difficult to assess, despite its critical role in understanding crack propagation. Future experimental efforts should not only focus on advanced microscopy and nano-indentation methods but also incorporate the use of high-energy X-rays, such as those available in synchrotron facilities. These high-energy X-rays can provide invaluable information about the bulk properties of materials, offering insights that are not attainable through conventional methods. This approach will significantly enhance our understanding of energy dynamics at the crack tip and facilitate the validation of computational models.

Incorporating energy-based concepts like Lagrangian or Hamiltonian mechanics into the study of fatigue and fracture in materials presents a promising yet largely unexplored avenue. These mechanics, known for their robust energy analysis framework, could transform our understanding of material behaviour under stress. Future research should focus on integrating these mechanics into material science models to deepen our theoretical understanding of energy dynamics within materials.

### 8.3. Energetic Modelling Procedure

A change in the numerical modelling of fracture and fatigue mechanics has been proposed, moving away from conventional models based on Newton’s theory and focusing on energy models. This new approach seeks not only to quantify the energy invested in the transformation of matter, but also to integrate the energy dissipated during the test, considering phenomena such as friction and molecular friction.

This energy approach finds its roots in Griffith’s theory, whose fundamental contribution was the introduction of the concept of the plastic radius, originally developed to explain the failure of brittle materials. It thus became a valuable theoretical framework for developing the relationship between theoretical and practical results.

The future directions in researching fatigue life prediction and fracture mechanics underscore the imperative to reduce the gap between macroscopic and microscopic approaches in material science. The challenge lies in developing universally applicable models that transcend diverse materials and loading conditions. Inspired by the successful integration of apparently divergent theories in fluid mechanics by expert Pradl, a similar rapprochement is sought between the theory of continuous media and particle systems in the fracture and fatigue domain. Overcoming challenges in experimental measurement and analytical modelling, particularly in accurately assessing energy within the plastic zone at the crack tip, necessitates advanced techniques such as high-energy X-rays available in synchrotron facilities. Future efforts should also explore the integration of energy-based concepts like Lagrangian or Hamiltonian mechanics, with their robust energy analysis framework, into material science models to revolutionise our understanding of material behaviour under stress. Additionally, a proposed shift in numerical modelling towards energy models, departing from conventional Newtonian-based models, holds promise in quantifying both the invested and dissipated energy during tests, with roots in Griffith’s theory providing a foundational framework for this innovative approach. This multifaceted strategy aims to advance our theoretical understanding and practical applications in the field of fatigue and fracture mechanics.

## Figures and Tables

**Figure 1 materials-16-07603-f001:**
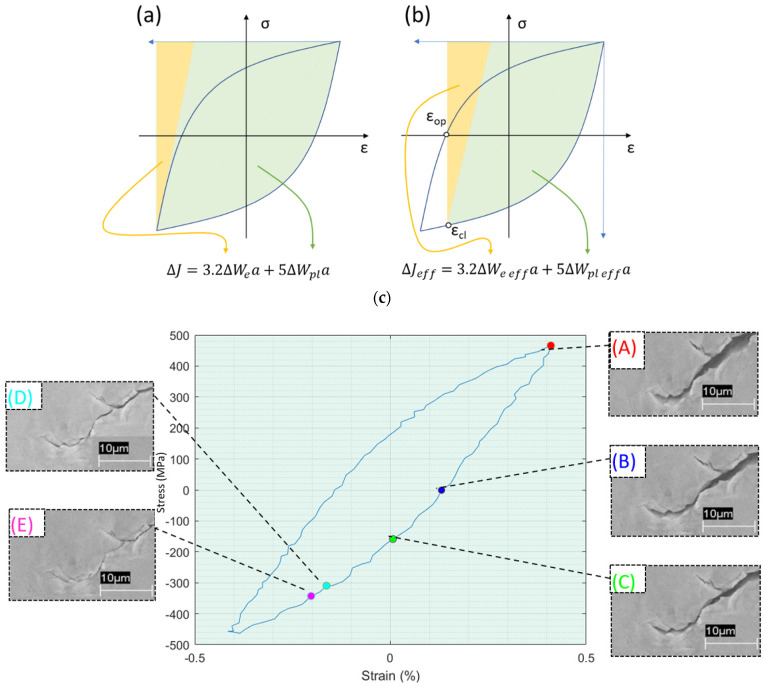
Schematic illustration of the procedure to determine ΔJ (**a**) and the ΔJ_eff_ (**b**). From [34]. (**c**) Stress–strain curve of a low cycle fatigue experiment at a plastic strain amplitude of about 0.3%. A–E present SEM micrographs showing the closing of the crack. From [34].

**Figure 2 materials-16-07603-f002:**
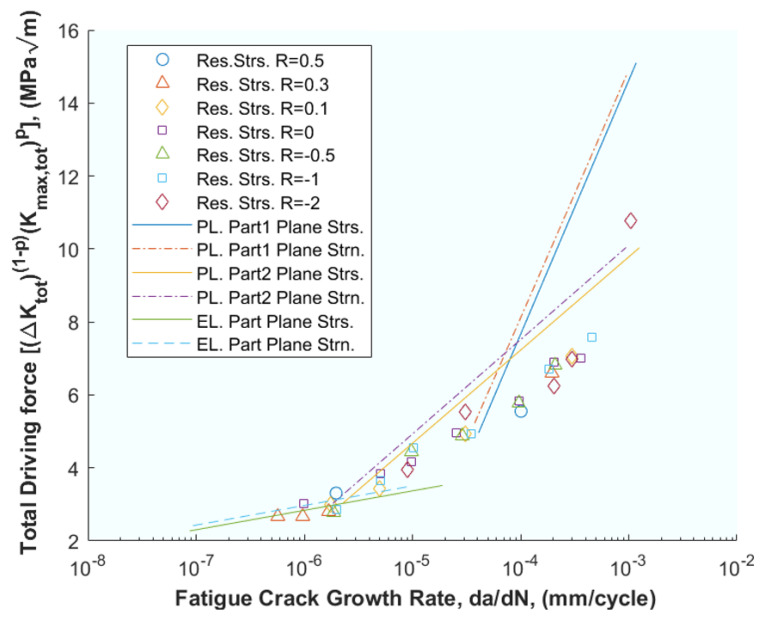
Fatigue crack growth data for Al 2024 T351 aluminium alloy as a function of the two-parameter driving force, ΔK. From [39].

**Figure 3 materials-16-07603-f003:**
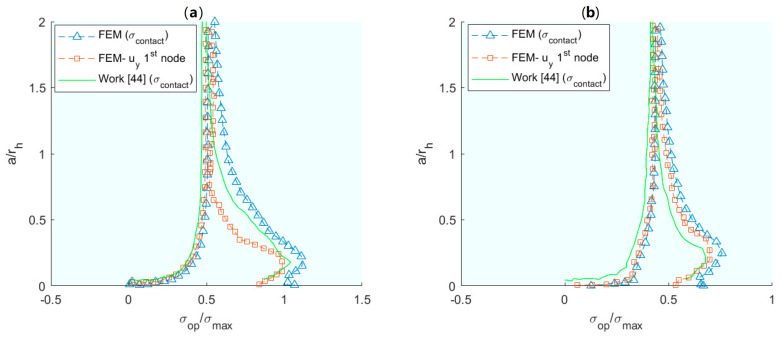
Opening stresses for an infinite plate with a circular hole and two radial symmetric cracks propagating under constant amplitude loading with and without residual stresses due to cold expansion (ur/r = 4%). (**a**,**b**) are for σ_0_/σ_max_ = 0.4 and 0.6, respectively, with R = 0. From [44].

**Figure 4 materials-16-07603-f004:**
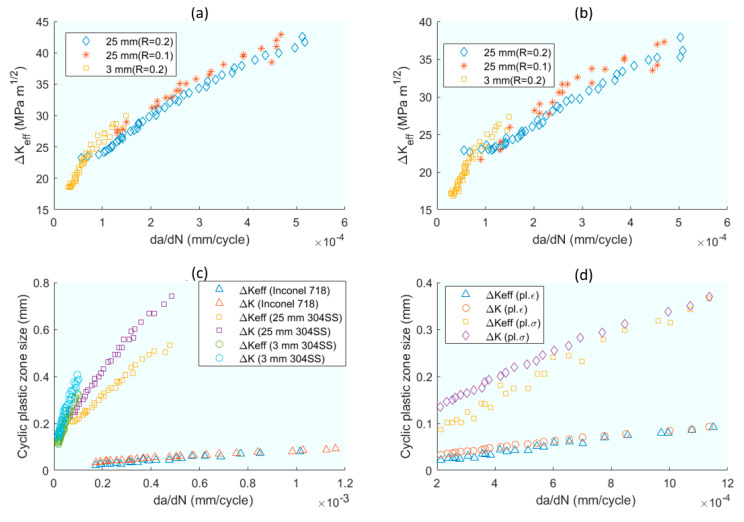
(**a**,**b**) Crack growth per cycle (da/dN) versus plastic CTOD range (ΔCTODp) for both tests for 304SS. (**c**) Comparison of FCGR specimens with relation to cyclic plastic zone plane strain conditions for Inconel 718 and 304SS. (**d**) Plane stress and strain for Inconel 718. From [58].

**Figure 5 materials-16-07603-f005:**
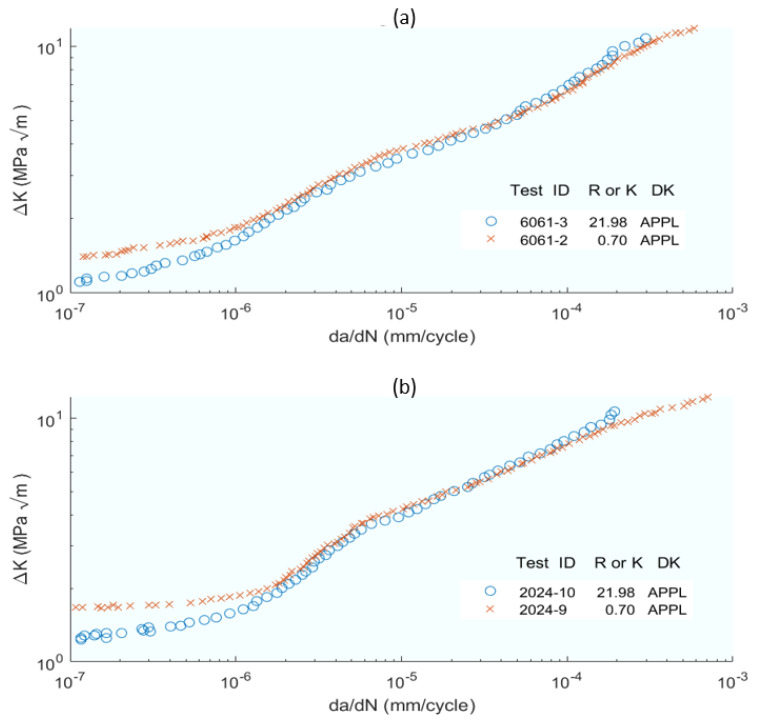
FCGR response of alloys (**a**) 6061−T6 and (**b**) 2024−T3 showing R = 0.7 data and constant K_max_ data (K_max_ = 22 MPa$\sqrt{m}$) as a function of K_app_. From [20].

**Figure 7 materials-16-07603-f007:**
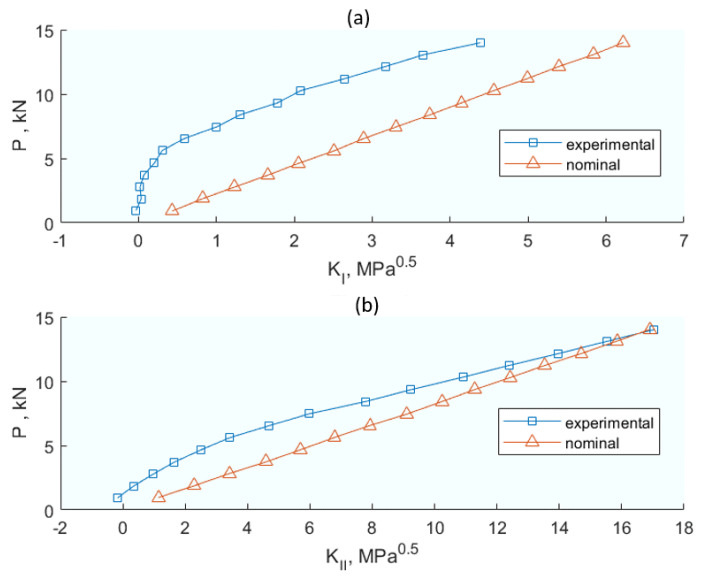
(**a**) Opening (K_I_) and (**b**) shearing (K_II_) mode stress intensity factors measured during the loading portion of a fatigue cycle (R = 0) with θ = 30º on a 7010 aluminium alloy centre-cracked plate. The nominal values were calculated neglecting closure effects. The experimental values were determined relative to an image captured at zero load. From [62].

**Figure 8 materials-16-07603-f008:**
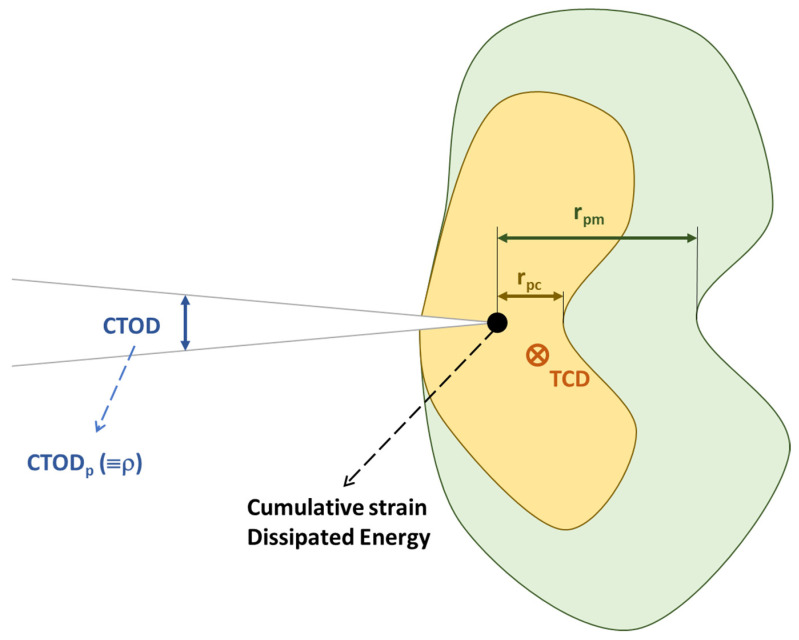
Interesting crack tip region/point, assuming that damage is a local issue—theory of critical distance (TCD). Region of reversed plastic zone *r_pc_* if fatigue damage accumulates. Region of forward plastic zone *r_pm_*.

**Figure 9 materials-16-07603-f009:**
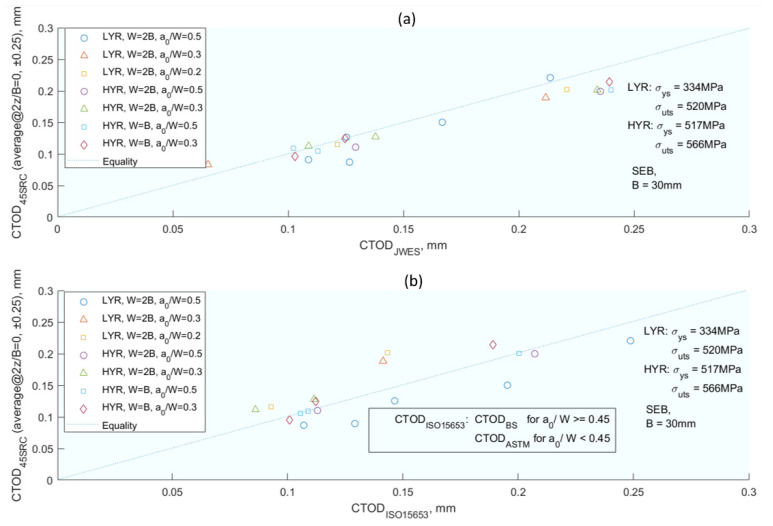
Correlation between CTOD calculated from P-Vg curve and actual CTOD at mid-thickness experimentally measured by silicone rubber casting for specimen with thickness B = 30 mm. (**a**) CTOD_JWES_. (**b**) CTOD_BS_. From [35].

**Figure 10 materials-16-07603-f010:**
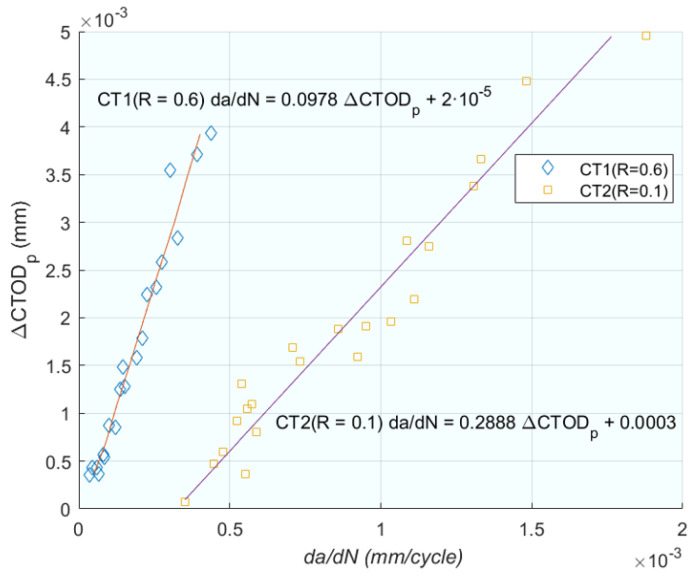
Crack growth per cycle (da/dN) versus CTOD range (ΔCTODp) for both tests).

**Figure 11 materials-16-07603-f011:**
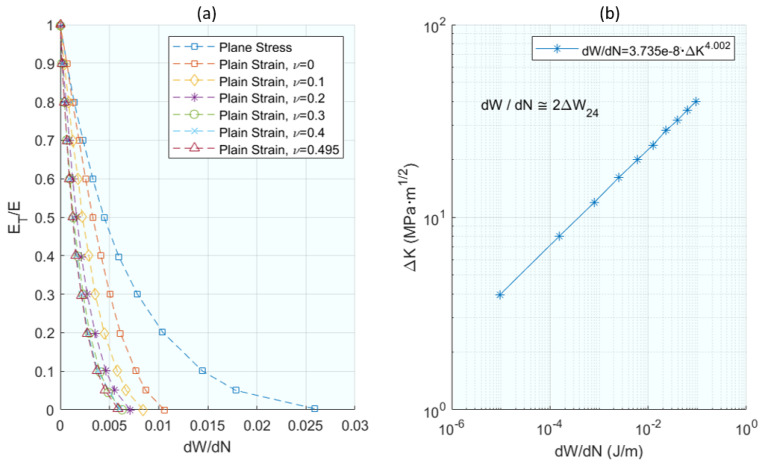
Evolution of fatigue crack propagation rate estimated by numerical simulation. (**a**) Master plot for R = 0 and (**b**) plastic dissipation per cycle vs. *ΔK* for a Titanium alloy. From [79].

**Figure 12 materials-16-07603-f012:**
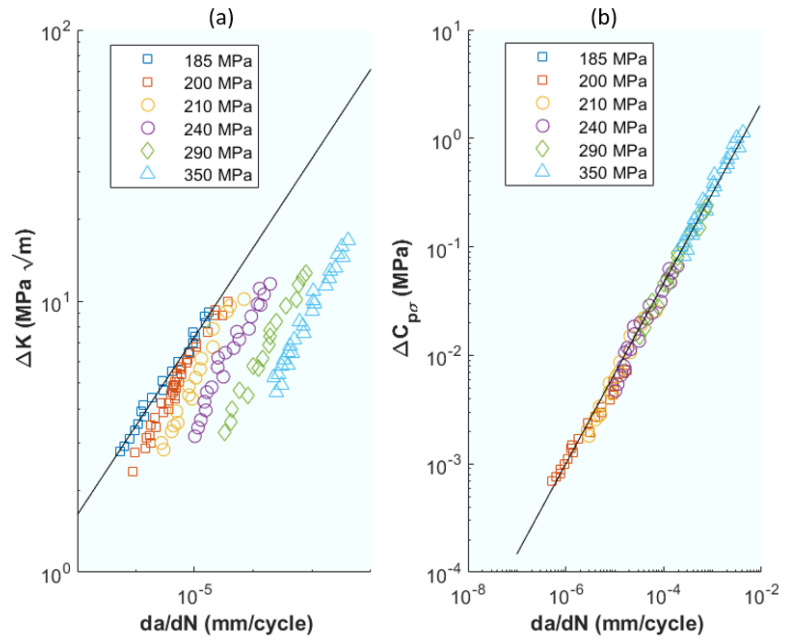
Fatigue crack growth data under stress control for cracks growing from starter holes in round tension specimens of 0.45% C steel, plotted in terms of (**a**) ΔK and (**b**) the change in net-section strain energy, ΔC_pσ_. From [80].

**Figure 13 materials-16-07603-f013:**
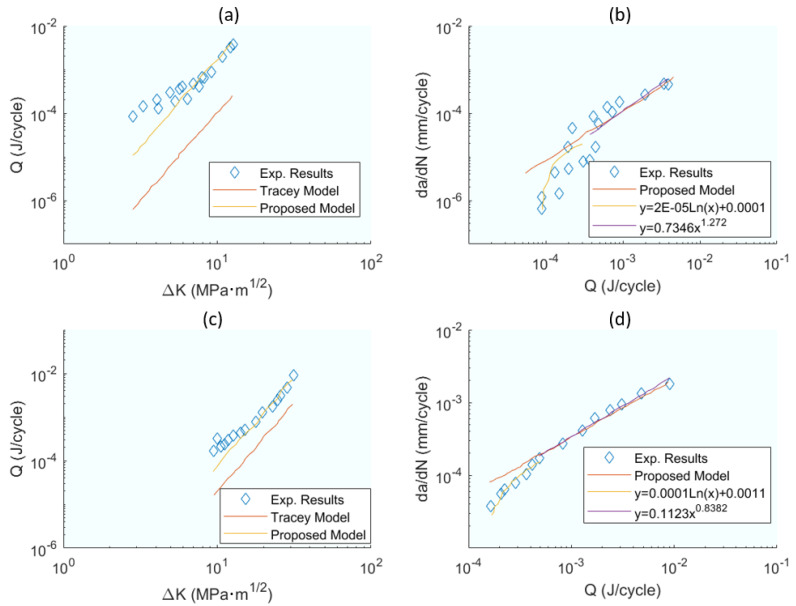
Comparison of measured and estimated dissipated energy per cycle for (**a**) 2024-T351 and (**b**) 7075-T7351. Comparison of measured evolution of da/dN with Q (**c**) for 2024-T351 (**d**) and 7075-T7351. From [83].

**Figure 14 materials-16-07603-f014:**
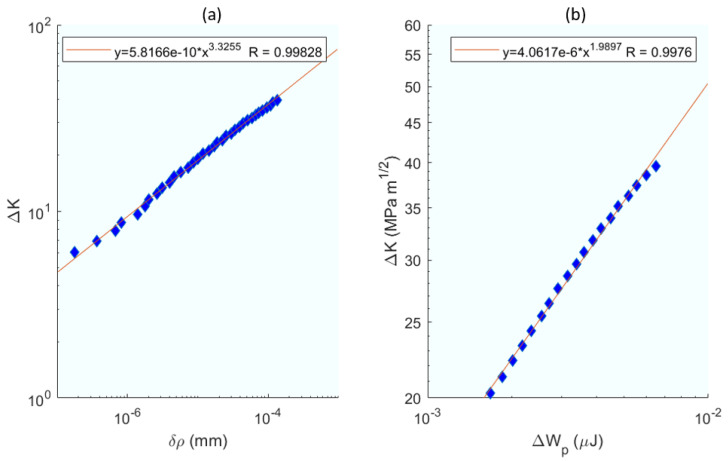
(**a**) The dependence of crack blunting parameter increment (per cycle) on the applied stress intensity factor, Δ*K*, that is described well by a power law relationship. (**b**) The dependence of crack tip plastic energy dissipation (per cycle) on the applied stress intensity factor, *ΔK*, that is described well by a power law relationship. From [84].

**Figure 15 materials-16-07603-f015:**
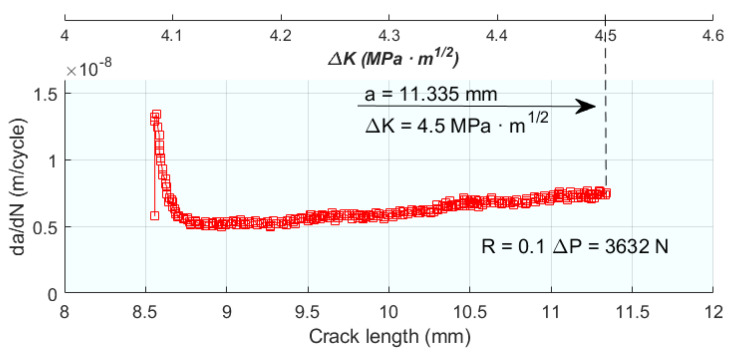
Evolution of fatigue crack propagation rate estimated by numerical simulation. From [87,88].

**Table 1 materials-16-07603-t001:** Damage parameters.

Damage Parameter	Equation
Stress	*σ_a_ = (σ′_f_ − σ_m_) (2N_f_) b*
Strain	*ε_a_ = (σ′_f_ − σ_m_) (1/E) (2N) fb + ε′f (2N_f_) c*
Energy	*ΔW_t_ = κ_t_ (2Nf) α^t^ + ΔW_0t_*
SWT	*ε_a_ σ_max_ = (σ′_f_)^2^ (1/E) (2N) f^2b^ + ε′_f_ (2N_f_) c^+b^*

**Table 2 materials-16-07603-t002:** CTOD and/or CTOD_P_ as a critical parameter used as a driving force in crack growth modelling. CTOD is Crack Tip Open Displacement, where _P_ denotes Plastic.

Ref.	Authors and Date	Description/Main Contribution	Methods	Sug.
[26]	Antunes et al., 2015	Analysis of remote compliance is the best numerical parameter to quantify the crack opening level. Establishment of an analytical relation between *CTOD* and *da*/*dN*. This relation was tested numerically.	Numerical	++
[35]	Kawabata et al., 2016	A new *CTOD* method is investigated considering the variation of crack tip blunting (strain hardening). The calculation formula is based on three-dimensional elasto–plastic FEM.	Numerical–Experimental	++
[43]	Shih 1986	Establish the relation between the J-integral and the crack opening displacement by exploiting the dominance of the Hutchinson–Rice–Rosengren singularity in the crack tip region.	Numerical–Experimental	++
[66]	Pokluda 2011	A discrete dislocation model of contact effects in small-scale yielding is presented. The model enables direct assessment of the magnitude of both plasticity and roughness-induced components of crack closure.	Analytical	++
[67]	Antunes et al., 2018	Establishment of a method of numerical calculation of the *CTOD* and *CTOD^P^* and their dependence on certain parameters.	Numerical	+++
[68]	Antunes et al., 2017	The 7050-T6 aluminium alloy cyclic plastic deformation was determined experimentally and modelled analytically. A 3D numerical model was developed to predict the *CTOD^P^*.	Numerical–Experimental–Analytical	+++
[70]	Antunes et al., 2018	First, experimental tests were conducted to obtain the relation between *CTOD* and FCGR. Then, numerical predictions of *CTOD^P^* were obtained for different crack lengths and *da*/*dN*.	Numerical–Experimental	+++
[71]	Tagawa et al., 2014	Numerical and experimental methods to determine a method to calculate *CTOD* and *CTOD^P^*.	Numerical–Experimental	+
[72]	Tagawa et al., 2009	The effects of *CTOD* testing methodologies on *CTOD* values were investigated according to tests conducted by the Japan Welding Engineering Society (WES).	Numerical–Experimental	++
[73]	Kayamori et al., 2010	Experimental investigations and analytical developments into crack tip opening displacement (*CTOD*) were conducted to establish the relationship between BS7448-CTOD and ASTM E1290-*CTOD*.	Numerical–Experimental	+
[74]	Kayamori et al., 2012	Two new CTOD calculations were proposed: for deep-notched specimens, a displacement–conversion CTOD; and for shallow-notched specimens, a J-conversion CTOD was proposed.	Numerical–Experimental	++
[75]	Vasco-Olmo et al., 2017	A methodology is developed to measure and analyse the CTOD and CTODP from experimental data.	Experimental	+++
[78]	Yates et al., 2010	The paper gives an overview of some DIC applications for crack tip characterisation such as CTOD and CTODP measures as well as data obtained.	Analytical–Experimental	++

**Table 3 materials-16-07603-t003:** Plastic size as a critical parameter used as driving force in the crack growth model.

Ref.	Specific Parameter	Authors and Date	Description/Main Contribution	Methods	Sug.
[26]	*R_pr_*	Antunes et al., 2015	Reverse plastic zone size. The crack closure phenomenon has a great influence on crack tip parameters, decreasing their values.	Numerical	++
[35]	*CWI*	Kawabata et al., 2016	Crack wake influence. A new factor f is introduced to correct the plastic term. In this factor, the blunted crack tip shape is considered to depend on the strain-hardening exponent, and f is given as a function of the yield-to-tensile ratio (*YR*) of the material and the specimen thickness.	Numerical–Experimental	+
[38]	ρ	Pommier and Risbet 2005	In the equations, special attention is paid to the elastic energy stored inside the crack tip plastic zone, sync. In practice, residual stresses at the crack tip are known to considerably influence fatigue crack growth.	Analytical	+++
[39]		Noroozi et al., 2005	The results demonstrate the crack closure influence on LCF behaviour. The change of crack closure from LCF to high cycle fatigue and their consequences for lifetime prediction.	Analytical–Experimental	++
[40]	*R_pr_*	Ould Chikh et al., 2008	Plastic zone size. The cyclic plastic strain can be the principal parameter for fatigue crack growth under a cyclic loading. Generally, FCGR is a plastic zone size *r_c_* function, and it increases as the plastic zone size increases.	Analytical	+++
[44]		De Matos et al., 2008	This paper shows that the residual stress field due to cold expansion has a strong influence on closure behaviour and therefore on fatigue crack propagation.	Numerical–Analytical	++
[56]	*εPA*	Borges et al., 2020	Fatigue crack growth (FCG) is simulated here by node release, which occurs when the accumulated plastic strain reaches a critical value.	Numerical	+++
[58]	*R_pc_*	Park et al., 1996	Plastic zone size experimental tests showed that plastic zone size was an important parameter in crack propagation.	Experimental	+++
[20]	*r_y_*	Donald and Paris 1999	The ACR and CWI methods measure the change in displacement at minimum load due to closure. That quantity is less subject to variability than the measurement of the opening load.	Analytical	+++
[61]	*da*/*dS*	Zhang et al., 2010	Parameter *da*/*dS* defines the fatigue crack propagation rate with the change of the applied stress at any moment of a stress cycle. The relationship between this new parameter and the conventional *da*/*dN* is given.	Numerical	++
[63]	*R_pc_*	Zhang et al., 2010	Plastic zone size. The results show that, near the crack tip, the reverse plastic zone size continues to change with a change in the applied compressive stress.	Numerical–Experimental	++
[66]	*PICC*	Pokluda 2011	Plasticity induced in the crack closure. There is a good qualitative agreement between the plasticity-induced shielding terms employed in the dislocation-based model and the continuum-based multi-parameter model.	Analytical	+
[67]	*R_pr_*	Antunes et al., 2018	Reverse plastic zone size. The increased crack plastic deformation also produces an increased crack closure phenomenon, which cancels the increased plastic deformation.	Numerical	++
[84]	ρ	Korsunsky et al., 2009	Plastic blunting, crack tip blunting. Two approaches were considered in the present study: the approach based on the consideration of crack tip blunting due to Pommier and Risbet [38], and the presently proposed approach based on the analysis of local energy dissipation in the immediate vicinity of the crack tip.	Analytical–Experimental	++

**Table 4 materials-16-07603-t004:** Energy as a critical parameter used as driving force in the crack growth model.

Ref.	Specific Parameter	Authors and Date	Description	Definition	Methodology
[67]	*CTOD_P_*	Antunes et al., 2018	Plastic energy dissipated per cycle	Plastic portion of the crack tip opening displacement. Plastic CTOD is obtained by subtracting the elastic CTOD from the total.	Numerical
[87,88]	*PM*	Zheng et al., 2013, 2014	Critical plastic energy, at the point close to the crack tip	The fatigue damage experienced by a point located within a specific distance from the crack tip can accurately represent the average damage condition at the crack tip area.	Numerical
[90]	*EC*	Kujawsky and Ellyin 1984	Critical plastic energy	Amount of plastic strain energy that a material can dissipate before experiencing fatigue failure.	Numerical
[91]	*EC*	Chalant and Remy 1983	Critical plastic energy	The strain gradient inside the grain at the crack tip.	Numerical
[94]	*ΔWt*	Branco et al., 2021	Critical plastic energy	Accounts for the mean stress effect, a measure of the energy dissipated per cycle. and is capable of unifying both the low-cycle and high-cycle fatigue regimes.	Numerical–Analytical
[93]	*χ_W_*	Zhu et al., 2018	Critical plastic energy	Reflects the distribution of both stress and strain gradients within the actual structure.	Analytical
[84]	*ΔW_p_*	Konsunsky et al., 2009	Equivalent deformation energy	Amount of energy dissipated due to plastic deformation at the crack tip during each loading cycle.	Analytical–Experimental
[79]	*ΔW_p_*	Klingbei 2003	Strain energy gradient	Change in total plastic dissipation per unit width during a specific cycle.	Numerical–Analytical
[80]	*U_L_*	Ravi Chandran 2018	Total dissipated plastic energy	Total net section strain energy in the crack plane of the ligament. Combination of elastic and plastic strain energies due to the increased stress.	Analytical–Experimental
[80]	*ΔC_pσ_*	Ravi Chandran 2018	Total dissipated plastic energy	Change in net section strain energy parameter in stress-controlled fatigue. *Equation (19)*	Analytical–Experimental
[80]	*ΔC_pε_*	Ravi Chandran 2018	Cumulative change in cyclic strain energy of the net section	Change in net section strain energy density in strain-controlled fatigue. *Equation (20)*	Analytical–Experimental
[81]	*U_pl_*	Quan and Alderliesten 2022	Plastic energy difference in the net section	Energy consumed in the process of crack growth through plastic deformation of the material surrounding the crack.	Numerical–Experimental
[81]	*U_e_*	Quan and Alderliesten 2022	Elastic energy difference in the net section	Variation in elastic strain energy stored throughout one full cycle of crack propagation.	Numerical–Experimental
[81]	*U_a_*	Quan and Alderliesten 2022	Surface energy difference in the net section	Surface energy differential dissipated through new crack surface formation.	Numerical–Experimental
[83]	*ΔW_p_*	Kheli et al., 2013	Stored deformation dissipation	Cyclic plastic strain energy, corresponding to one loading cycle. *Equation (21)*	Numerical–Experimental
[83]	*U*	Kheli et al., 2013	Dissipation new crack surface formation	Specific energy, energy dissipated per unit volume during fatigue crack growth. *Equation (22)*	Numerical–Experimental
[83]	*Q*	Kheli et al., 2013	Plastic energy of the hysteresis cycle characteristic of cyclic loads	Total dissipated energy in the specimen during fatigue crack growth. *Equation (23)*	Numerical–Experimental

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
