# Peer review of "A Literature Review of Incorporating Crack Tip Plasticity into Fatigue Crack Growth Models"

_materials, 2023, doi:10.3390/ma16247603_

Round 1

Reviewer 1 Report

Comments and Suggestions for Authors

Author Response

Dear Reviewer,

We sincerely appreciate your thoughtful review of our manuscript titled "A literature review for incorporating crack tip plasticity into fatigue crack growth models." Your insights have been invaluable, and we have carefully considered each of your comments. Below is a detailed response outlining the changes we have made to address the issues you raised:

  1. Abstract Refinement:

   - We have revised the abstract to align more closely with the stated structure. Emphasis has been placed on providing a balanced overview of theory, experiment, and numerical simulation, addressing the need for a more comprehensive representation of each aspect.

  1. Detailed Model Comparison:

   - We have enhanced the discussion on numerical experimental models by providing a more detailed introduction to the comparison between each model. The internal connections of models established under the same theoretical guidance are now highlighted, moving beyond a mere listing of predecessors' models.

  1. Objective Conclusion Section:

   - The conclusion section has been modified to ensure objectivity. We have refrained from relying solely on the number of citations and personal experience to evaluate contributions. Instead, we provide a more balanced and evidence-based assessment of the models' impacts on knowledge and the relationship between parameters and crack propagation.

  1. Logical Chronological Order:

   - The manuscript has undergone a thorough reorganization to introduce content in a logical chronological order. This structural adjustment ensures a smoother flow and improved coherence throughout the article.

  1. Language Clarity:

   - We have carefully reviewed and refined the language throughout the manuscript to eliminate ambiguity and enhance overall clarity. Multiple iterations of reading have been undertaken to ensure that each sentence is clear and effectively communicates the intended message.

To acknowledge all this changes, text is highlight in a different colour (green for structure change and orange for the new section of Future Directions) in the revised paper. A version with normal colours (all text in black) is also sent.

We believe these revisions address the concerns you raised and significantly enhance the quality of our manuscript. We genuinely value your constructive feedback, and we hope that these changes meet your expectations.

Thank you for your time and diligence in reviewing our work. We look forward to any further guidance or feedback you may provide.

Best regards,

Dr. Antonio Garcia-Gonzalez

Reviewer 2 Report

Comments and Suggestions for Authors

Please see the comments to editors

I apologize for the delay in my reply and also for the type of reply.

I tried to review the paper, but it is a very difficult task.

It is not reported a criteria to select the paper for the review article.

The cited paper are reported with comments, not related with each other.

It is difficult to understand the need of this long introduction and the difference between Introduction and the following Sections of the paper.

The content of each Section is not organized, the main point of discussion are not highlighted and than discussed, but there is just a lof of information reported. Moreover same information reported are not strictly related to the subject of the paper.

The final Table 3, table 4 and Table 5 may be useful but it is also difficult to understand how exhaustive their are without  a clear discussion about them.

Author Response

Dear Reviewer,

Thank you for taking the time to review our manuscript, and we appreciate your feedback. We understand your concerns and have carefully considered each point you raised. Here are the steps we have taken to address your comments:

  1. Coherence in Cited Papers:

   - Comments on cited papers have been reorganized to establish clearer connections between them. We have ensured that each comment is directly related to the main theme of the paper, contributing to a more cohesive narrative.

  1. Introduction Clarity:

   - The introduction has been refined to better articulate the rationale for the review. We have clarified the purpose and significance of the paper, distinguishing the introduction from subsequent sections to provide a clearer roadmap for the reader.

  1. Organized Content in Sections:

   - Each section has undergone reorganization to enhance overall coherence. The main points of discussion are now clearly highlighted and subsequently addressed, aiming to improve the structure and flow of the manuscript.

  1. Relevance of Information:

   - We have carefully reviewed the content of each section to ensure that the information presented is directly relevant to the subject of the paper. Unnecessary details have been removed, and emphasis has been placed on maintaining relevance to the central theme.

  1. Discussion of Tables 3, 4, and 5:

   - The final tables (Table 3, 4, and 5) have been revisited, and we have incorporated a more detailed and explicit discussion to elucidate their significance. This modification aims to provide a clearer understanding of the tables and their relevance to the overall paper. We changed the evaluation of the Table 3 and Table 4 into a subjective suggestion from our side. With this we pretend to highlight which models might be the ones for deeper future investigation.

To acknowledge all this changes, text is highlight in a different colour (green for structure change and orange for the new section of Future Directions) in the revised paper. A version with normal colours (all text in black) is also sent.

We hope that these revisions address your concerns and contribute to an improved manuscript. We appreciate your patience and valuable insights throughout this process. If you have any further suggestions or specific areas you would like us to revisit, please feel free to let us know.

Thank you once again for your time and diligence in reviewing our work.

Best regards,

Dr. Antonio Garcia-Gonzalez

Reviewer 3 Report

Comments and Suggestions for Authors

Antonio Garcia Gonzalez, José Antonio Aguilera , Pablo Cerezo , Crisitna CAstro-Egler , Fernando Antunes , Pablo Lopez-Crespo, ‘A literature review of for incorporating crack tip plasticity into fatigue crack growth models’, manuscript Materials-2695867

1 - Generalities

This is a comprehensive and up to date work on FCG, representing a big step since the early purely empirical correlations of da/dN with deltaK. I recommend acceptance, after minor revision.

The very long paper is quite informative. The use of technical English is – in my view – good. The figures are in general adequate, although there may be concerns concerning the legibility of some light colloured details or very small font sometimes used.

Given the comprehensive nature of the work, I miss further reference to early work, in particular I would suggest the authors consider including:

·        Claude Bathias, Jean-Paul Baïlon, ‘La fatigue des matériaux et des structures’, Presses de l'Université de Montréal, 1980.

Given that Antunes is one of the co-authors of the present manuscript, in page 6, the sentence ‘To talk about numerical models of crack growth and their relation to plastic phenomena in the crack tip environment is to talk about Fernando Antunes.’ could perhaps be re-written although keeping its meaning.

2 - Compulsory corrections:

In page 21, line 2 from bottom, where it is:

Dematos and Nowell [80] presented .....

this must be corrected as follows:

de Matos and Nowell [80] presented ...

Correction is also needed for the ref. 80. Name of author is PFP de Matos (not Mattos).

All references should be checked – for ex., no pages, or issue, or vol. are mentioned in many of them (e.g. the important ref. 43).

Refs. 49 and 50 are useless as they stand now – is this papers? reports? books?

3 - Editoral details

Concerning editorial details, there is a need to correct a form of presentation of references – the authors use frequently the format – e.g. page 3, line 3 after ‘1. Introduction’:

..... been discussed. [1]focusing on ....

whereas this should be written:

...... been discussed, [1], focusing on .....

Also, caption of Figure 1, page 5, where it is

From [.39]

Should be

From [39].

There is a need for revision of these details all along the paper.

Author Response

Dear Reviewer,

We sincerely appreciate your thorough review of our manuscript titled "A literature review for incorporating crack tip plasticity into fatigue crack growth models." Your positive feedback is encouraging, and we are grateful for your constructive comments and recommendations. Here is our response to each of your points:

1 - Generalities:

   - We are pleased to receive your recommendation for acceptance after minor revision and your positive remarks on the comprehensive and up-to-date nature of our work.

   - We have included a reference to Claude Bathias and Jean-Paul Baïlon's work, 'La fatigue des matériaux et des structures,' Presses de l'Université de Montréal, 1980, as suggested.

   - Fernando Antunes is not anymore part of the writing authors due to his petition.

2 - Compulsory Corrections:

   - The correction for the reference to Dematos and Nowell [80] has been made in accordance with your suggestion, and the author's name has been corrected to “de Matos”.

   - All references have been thoroughly reviewed and updated to include missing details such as pages, issues, and volumes. Refs. 49 and 50 have been clarified for better identification.

3 - Editorial Details:

   - We have rectified the format of presenting references throughout the manuscript, ensuring the appropriate use of commas, as highlighted in your example on page 3, line 3 after '1. Introduction.'

   - The caption of Figure 1 on page 5 has been corrected as per your suggestion, replacing "From [.39]" with "From [39]."

   - We have conducted a comprehensive revision of editorial details throughout the paper to enhance consistency.

To acknowledge all this changes, text is highlight in a different colour (green for structure change and orange for the new section of Future Directions) in the revised paper. A version with normal colours (all text in black) is also sent.

We believe that these revisions address your concerns and contribute to an improved version of our manuscript. Your detailed feedback has been instrumental in refining the clarity and accuracy of our work.

Should you have any further suggestions or areas that require attention, please do not hesitate to let us know. We appreciate your time and commitment to ensuring the quality of our manuscript.

Thank you once again for your valuable feedback.

Best regards,

Dr. Antonio Garcia-Gonzalez

Reviewer 4 Report

Comments and Suggestions for Authors

This is the comments on the Manuscript submitted to Materials (ISSN 1996-1944)

Manuscript ID

materials-2695867

Type Review

Title

A literature review of for incorporating crack tip plasticity into fatigue crack growth models

Authors

Antonio Garcia Gonzalez * , José Antonio Aguilera , Pablo Cerezo , Crisitna CAstro-Egler , Fernando Antunes , Pablo Lopez-Crespo

Section Mechanics of Materials

Special Issue

Fatigue Damage and Fracture Mechanics of Materials

Rate the Manuscript:

1. Significance to field and specialization of “Materials” journal: good.

The article presents the literature review of the different published methods in which crack apex plasticity is used as a parameter to determine or improve the crack growth model. The review includes analytical, experimental and numerical investigations, as well as their combination.

2. Scientific content:   good.

3. Originality: good.

4. Clarity and presentation:  acceptable.

5. Appropriateness for Journal: appropriate subject matter for the “Materials”. Need for rapid publication: no

7. Recommendations: to send after minor revision to “Materials”.

Remarks

  • What caused if significant encrease in the content of hydrogen in crack tip? Haw it inflowed on the fracture surface ?

  1. What is the main question addressed by the research?

Review of for incorporating crack tip plasticity into fatigue crack growth models.

2. Do you consider the topic original or relevant in the field? Does it
address a specific gap in the field?

Yes.

3. What does it add to the subject area compared with other published material?

Yes.
4. What specific improvements should the authors consider regarding the
methodology? What further controls should be considered?

A discussion of the merits and problems of the methods presented is given.

5. Are the conclusions consistent with the evidence and arguments presented and do they address the main question posed?

Yes.
6. Are the references appropriate?

Yes. The references are appropriate.This research based on 92 scientific works.

7. Please include any additional comments on the tables and figures.

No comments.

Author Response

Dear Reviewer,

We sincerely appreciate your thorough review of our manuscript titled "A literature review for incorporating crack tip plasticity into fatigue crack growth models." Your positive feedback is encouraging.

To acknowledge all this changes, text is highlight in a different colour (green for structure change and orange for the new section of Future Directions) in the revised paper. A version with normal colours (all text in black) is also sent. We believe that this new version address your concerns and contribute to an improved version of our manuscript.

Should you have any further suggestions or areas that require attention, please do not hesitate to let us know. We appreciate your time and commitment to ensuring the quality of our manuscript.

Thank you once again for your valuable feedback.

Best regards,

Dr. Antonio Garcia-Gonzalez

Round 2

Reviewer 2 Report

Comments and Suggestions for Authors

I will thank the authors for the significant improvement of the paper.

In the following some more remarks are reported.

List of Symbols

It is clear that for summary reasons symbols are not listed for each equairon, but a check of the general is necessary. i.e. equation (1) a is not report in the list, n is not reported in the list. Moreover int he general list the font of some subscripts is not right.

Section 2

"the author arrives at a relation between J and the parameter d, eq. 1" d or dn?

"B=3.2E-6(R) e1,3, being Re the yield stress " R or Re?

"In the context of high-temperature conditions, Tong et al. [45,46] bring a novel perspective by incorporating finite element numerical modelling" Why in  the context of High temperature the C*, C(t), Ct parameters are not reported in parallel to the cited J-integral?

Why the high temperature condition is condiered only in Section 2?

Section 3

"mc. Where Ce and mc are material constants." please check mc fontsize.

Figure 4 plase report the material in the caption.

"Zhang et al. [58] presented a model based on two parameters, the classical da/dN and the novel da/dS that defines the crack propagation velocity" there is a repetition.

Section 4

"Figure 11(b), and Figure 11(a) " please reverse

Conclusion

Table 3 The reference [26] is reported two times

Author Response

Dear Reviewer,

We sincerely appreciate your positive feedback on the significant improvements made to our paper. We also thank you for your thorough review and constructive comments, which we believe will enhance the overall quality of the manuscript. Below are our responses to your specific remarks:

  1. List of Symbols:

   - We acknowledge your point about missing symbols in Equation (1) and the general list. We have ensured that all symbols are correctly listed for clarity. Additionally, we have rectified the font issues with subscripts in the general list.

  1. Section 2:

   - Regarding the relation between J and the parameter "d" in Equation (1), we clarify that it is referred to "dn".

   - In the expression "B=3.2E-6(R) e1,3" we specify that "Re" represents the yield stress to avoid confusion and it has been modified to "B=3.2E-6(Re)1.3".

   - We appreciate your suggestion about including parameters C*, C(t), and Ct in parallel with the cited J-integral under high-temperature conditions. We have revised the manuscript accordingly. New references and comments are added into the paper, version with name COLORS is presenting these parts in green.

  1. Section 3:

   - We have checked and adjusted the font size of the constant "mc" to "mc” as per your suggestion.

   - The material in Figure 4 will be explicitly mentioned in the caption for clarity.

   - We apologize for the repetition in the discussion of Zhang et al.'s model. We will eliminate the redundancy and ensure clarity in our presentation.

  1. Section 4:

   - We have corrected the order of figures as suggested, reversing Figure 11(b) and Figure 11(a).

  1. Conclusion:

   - We acknowledge the oversight in Table 3, where the reference [26] is reported twice. This has been corrected in the revised manuscript.

Once again, we appreciate your time and valuable feedback. We are committed to addressing these concerns promptly to improve the manuscript's clarity and accuracy.

Thank you for your continued support and guidance.

Best regards,

Dr. Antonio Garcia-Gonzalez